# AdaWaveNet: Adaptive Wavelet Network for Time Series Analysis

**Han Yu**                                                                    *hy29@rice.edu*
*Department of Electrical and Computer Engineering*
*Rice University*

**Peikun Guo**                                                               *pg34@rice.edu*
*Department of Computer Science*
*Rice University*

**Akane Sano**                                                         *akane.sano@rice.edu*
*Department of Electrical and Computer Engineering*
*Rice University*

**Reviewed on OpenReview:** *https://openreview.net/forum?id=m4bE9Y9FlX*

## Abstract

Time series data analysis is a critical component in various domains such as finance, healthcare, and meteorology. Despite the progress in deep learning for time series analysis, there remains a challenge in addressing the non-stationary nature of time series data. Most of the existing models, which are built on the assumption of constant statistical properties over time, often struggle to capture the temporal dynamics in realistic time series and result in bias and error in time series analysis. This paper introduces the Adaptive Wavelet Network (*AdaWaveNet*), a novel approach that employs Adaptive Wavelet Transformation for multi-scale analysis of non-stationary time series data. *AdaWaveNet* designed a lifting scheme-based wavelet decomposition and construction mechanism for adaptive and learnable wavelet transforms, which offers enhanced flexibility and robustness in analysis. We conduct extensive experiments on 10 datasets across 3 different tasks, including forecasting, imputation, and a newly established super-resolution task. The evaluations demonstrate the effectiveness of *AdaWaveNet* over existing methods in all three tasks, which illustrates its potential in various real-world applications. The code implemented for the *AdaWaveNet* is available at https://github.com/comp-well-org/AdaWaveNet.

## 1 Introduction

Time series data, extensively encountered in various domains, including finance, healthcare, and meteorology, require effective analytical methodologies (Esling & Agon, 2012). Therefore, understanding and analyzing time series data has triggered substantial interest in various real-world applications. Recently, the rapid development of deep learning has significantly transformed the landscape of time series analysis. These advancements have triggered breakthroughs in various applications, including forecasting (Zhou et al., 2021; 2022b; Wu et al., 2021), imputation (Wu et al., 2022; Xu et al., 2022), and anomaly detection (Blázquez-García et al., 2021; Li & Jung, 2023).

However, even with the promising performances of the aforementioned methods, a notable limitation in this research area is the inadequate focus on the non-stationary nature of time series data. Non-stationarity, with its evolving statistical properties and time-dependent patterns, poses a significant challenge for traditional deep learning models (Hyndman & Athanasopoulos, 2018; Shumway et al., 2017). The constantly shifting nature of time series data can make it difficult for the aforementioned methods to fully capture its dynamic patterns, which potentially leads to inaccurate analysis.

To address the non-stationary challenge, recent efforts have aimed to adapt deep learning methods for temporal dynamic analysis (Liu et al., 2022b; 2023b). However, due to the designing basis, such as instance-wise normalization and Fourier transform, these methods may lack the adaptability to process multi-scale features and capture the changing temporal dynamics across different signals. Thus, despite the advancements in modeling non-stationary time series data by these methods, there remains a critical need for an approach that combines multi-scale analysis with efficient computational strategies and adaptability.

In response to these challenges, we propose *AdaWaveNet*, a novel architecture that employs adaptive wavelet transformations within an efficient, multi-scale framework. Unlike existing wavelet-based methods such as FEDformer (Zhou et al., 2022b), which relies on manually tuned wavelet parameters and a fixed Transformer architecture, *AdaWaveNet* leverages the lifting scheme (Sweldens, 1998) to enable fully learnable, adaptive wavelet transformations. Also, expanded from the existing time-series decompositon, we proposed a grouped linear module to enhance the dependencies among varying channels. This approach dynamically adjusts to the data's temporal properties, which offers superior flexibility and robustness in handling non-stationary time series. By combining adaptive multi-scale analysis with an efficient computational design, *AdaWaveNet* addresses critical gaps in current methods for analyzing complex time series data.

Our contribution can be summarized as:

- We introduce *AdaWaveNet*, a novel architecture designed to address the challenges posed by non-stationary time series data through an adaptive, multi-scale approach. Differing from prior decomposition-based or fixed-parameter methods, *AdaWaveNet* leverages a learnable wavelet transformation via the lifting scheme, which enables dynamic adaptation to evolving statistical properties within the data. While we leverage time-series decomposition module that separates trend and seasonal components following previous methods, the adaptability is further enhanced by a grouped linear module, which uniquely captures channel-specific dependencies.

- We establish a new benchmark for super-resolution in the field of time series data. This benchmark aims to enhance the quality of data obtained from under-sampled sequences and improve the overall efficiency and effectiveness of time series data monitoring and analysis.

- Our extensive evaluations demonstrate that *AdaWaveNet* outperforms existing methods in forecasting and super-resolution tasks. These results suggest the capability of the proposed method for diverse real-world applications.

## 2 Related Work

### 2.1 Time Series Analysis with Deep Learning

The evolution of deep learning has significantly impacted temporal modeling and time series analysis. Recurrent neural networks (RNNs), such as those based on Long Short-Term Memory (LSTM) (Hochreiter & Schmidhuber, 1997; Siami-Namini et al., 2019), are designed to capture temporal dependencies through internal states. Multi-Layer Perceptrons (MLPs) (Zeng et al., 2023; Li et al., 2023) have shown effectiveness in temporal modeling by processing point-wise projections of sequences. Convolutional Neural Networks (CNNs) (Lea et al., 2017; Liu et al., 2022a) excel in extracting hierarchical features and detecting complex patterns, leveraging their strength in spatial and temporal data processing. Moreover, the Transformer variants have demonstrated remarkable results in time series applications by capturing long-range dependencies and processing entire sequences efficiently (Zhou et al., 2022b; Liu et al., 2023a; 2022b; Zhang & Yan, 2022). Nevertheless, in real-world applications, these well-proven structures may struggle with non-stationarity in time series data because their learned patterns and dependencies are based on the assumption of consistent statistical properties. On the other hand, most of the realistic temporal data, which is non-stationary with dynamics over time, violates the consistent assumption.

These methods have been developed and applied to various tasks, including forecasting (Zhou et al., 2021; 2022b; Wu et al., 2021), imputation (Wu et al., 2022; Xu et al., 2022), and anomaly detection (Blázquez-García et al., 2021; Li & Jung, 2023). However, the field of super-resolution in time series analysis remains relatively

unexplored. This technique, crucial for enhancing signal quality and detail, can significantly benefit sensing applications. For example, in wearable sensors, super-resolution can extend battery life and reduce storage needs by enabling post-processing enhancement of data resolution instead of continuous high-frequency sampling. This approach not only conserves resources but also provides detailed insights for precise tasks such as health monitoring.

### 2.1.1 Decomposition-Based Methods

Decomposition-based methods have emerged as promising techniques for time series analysis by separating data into interpretable components such as trend and residual terms, which enhances the ability to capture both short-term and long-term patterns (Wu et al., 2021; Wang et al., 2024; Zhu et al., 2023; Cao et al., 2023; Hu et al., 2023). For instance, DRCNN (Zhu et al., 2023) uses a multi-resolution approach with convolutional kernels to improve forecasting accuracy across different time scales. Similarly, MICN (Wang et al., 2023) combines multi-scale local and global context modeling through convolution and isometric convolutions, achieving significant improvements in long-term forecasting tasks. TEMPO (Cao et al., 2023) extends Transformer-based models by introducing a prompt-based generative pre-training architecture that decomposes time series into trend, seasonal, and residual components and establishes a foundational model for time series forecasting.

Despite the success of the aforementioned decomposition-based methods, they have limitations in handling the dynamic and evolving nature of non-stationary time series data. For instance, DRCNN's focus on fixed decomposition into residual and trend terms may fail to capture complex temporal shifts over time, while MICN, though addressing both short-term and long-term patterns, relies on pre-defined convolutional kernels that lack adaptability to dynamically changing signals. To leverage the benefits of decomposition while incorporating adaptability, our proposed *AdaWaveNet* introduces a novel adaptive wavelet-based lifting scheme that dynamically learns wavelet coefficients through end-to-end training.

### 2.2 Non-Stationarity-Enhanced Models

Recent developments in time series analysis have started addressing non-stationarity issues (Liu et al., 2022b; Zhou et al., 2022b; Liu et al., 2023b;c). For instance, Liu et al. (2022b) introduced Non-stationary Transformers with strategies like *Series Stationarization* and *De-stationary Attention* to standardize signal statistics over time. Zhou et al. (2022b) employed frequency domain-enhanced attentions in Transformers, incorporating Fourier and wavelet transform-based techniques. Additionally, Liu et al. (2023b) integrated Koopman operator theory for analyzing non-linear dynamical systems by transforming signals into a linear, high-dimensional space. While these methods effectively model stationarity, they exhibit limitations such as the need for manually tuned filters in FEDformer and extensive computations for long-term signals or high-dimensional projections.

Recent advancements in adaptive wavelet-based models, such as the Adaptive Multi-Scale Wavelet Neural Network (AMSW-NN) (Ouyang et al., 2021), have demonstrated the effectiveness of combining multi-scale convolutional neural networks with depthwise convolutions. However, AMSW-NN primarily focuses on generating candidate frequency decompositions without explicitly considering inter-channel relationships or dynamic adaptability at different scales. To address these gaps, we propose the Adaptive Wavelet Network (*AdaWaveNet*). AdaWaveNet utilizes a novel lifting scheme that learns adaptive wavelet coefficients through end-to-end training and incorporates a grouped linear module for efficient trend processing. This dual-residual approach dynamically captures both fine-grained and broad patterns, which enhances adaptability and performance across diverse time series datasets.

## 3 Background

Non-stationary time series data often contain transient behaviors and changing frequencies, which makes traditional methods that assume constant statistical properties less effective. Wavelet analysis addresses this by enabling multi-resolution decomposition, where each scale handles different frequency bands over time. For instance, high-frequency components can capture rapid changes or anomalies that occur over a short

duration; whereas low-frequency components can capture long-term trends or patterns that evolve more gradually. Thus, wavelet transforms provide a natural way to represent non-stationary data by focusing on both time-localized features and frequency dynamics, which makes them particularly useful for time series where the underlying statistical properties are not consistent.

This section reviews the wavelet transform and the lifting scheme (Sweldens, 1998) concepts to provide foundational knowledge essential for understanding the proposed method.

### 3.1 Wavelet Transform

The wavelet transform is a versatile tool for analyzing signals across multiple scales, simultaneously capturing both frequency and temporal information. Unlike the Fourier transform, which focuses exclusively on the frequency domain, the wavelet transform enables a time-frequency analysis by decomposing a signal into components that reveal its structure at different resolutions. This ability is crucial for handling non-stationary time series, which often contain short-term fluctuations (high-frequency) alongside long-term trends (low-frequency). Given a signal $f$, the discrete wavelet transform decomposes it into the following form:

$$f(x) = \sum_{i,j} \langle f, \psi_{i,j}(x) \rangle \psi \left( 2^i x - j \right) = \sum_{i,j} w_{i,j} \psi_{i,j}(x)) \tag{1}$$

In the equation, $\psi_{i,j}(x) = \psi \left( 2^i x - j \right)$ represents the wavelets at different scales $i$ and translations $j$, with $w = \langle f, \psi(x) \rangle$ denoting the wavelet coefficients.

The wavelet transformation provides an extensive capability for analyzing the changing dynamics and non-stationarity in time series data, which shows advances compared to the Fourier transform's frequency-centric approach. Traditional wavelet bases, such as Haar (Haar, 1909), Daubechies (Daubechies, 1988), and Biorthogonal (Cohen et al., 1992) wavelets, have been widely used in various real-world applications. However, the selection of an optimal wavelet basis remains a challenge, as the effectiveness of different bases can vary considerably depending on the specific characteristics and structure of the real-world time series data being analyzed.

### 3.2 Lifting Scheme

The lifting scheme, also known as the second-generation wavelet approach, was introduced by Sweldens (1998) to enhance the flexibility and adaptability of wavelet transforms. Differing from traditional wavelets built from dilations and translations of a single function, the lifting scheme constructs wavelets in the spatial domain using a series of simple operations. This approach preserves essential wavelet properties while offering greater adaptability to specific signal characteristics.

The scheme processes an input signal $x$ to segregate it into approximation (c) and detail (d) sub-bands, which achieves multi-resolution analysis through three main stages: split, update, and predict. This method enables the creation of custom wavelets that can be more efficiently computed and better adapted to irregular data structures or non-standard sampling grids. By doing so, the lifting scheme provides an efficient route to achieve similar mathematical properties and practical results as traditional wavelets, which makes it particularly valuable for various signal processing applications.

The procedures of the lifting scheme can be considered as:

**Split**: The input signal is divided into two non-overlapping components: the even $(x_e)$ and odd $(x_o)$ components, denoted as $x_e[n] = x[2n]$ and $x_o[n] = x[2n + 1]$.

**Update**: This stage separates the signal in the frequency domain to generate the approximation $c$. An update operator $U(\cdot)$ is applied to a sequence of neighboring odd polyphase samples, yielding $c[n] = x_e[n] + U(x_o^{L_U}[n])$.

**Predict**: Given the correlation between $x_e$ and $x_o$, a predictor $P(\cdot)$ is developed for one partition based on the other. The detail sub-band $d$, is computed as the prediction residual $d[n] = x_o[n] - P(c^{L_P}[n])$.

The lifting scheme improves the flexibility of wavelet transformations by allowing for a data-driven adaptation of wavelet coefficients, which makes it more suitable for analyzing non-periodic and intricate signals frequently

encountered in real-world applications. Also, in the proposed *AdaWaveNet* architecture (Section 4), the lifting scheme is crucial for enabling a hierarchical decomposition of the time series data, where the signal is split into coarser approximations and finer details at each level. This allows the model to capture both local fluctuations and global trends in the data and model the non-stationary characteristics of time series.

# 4 Adaptive Wavelet Network

We propose an Adaptive Wavelet Network (*AdaWaveNet*), which comprises a time series decomposition module, stacked adaptive wavelet (*AdaWave*) blocks based on the lifting scheme (Sweldens, 1998), and a grouped linear module. Figure 1 illustrates the overall framework of our method. We denote the input sequence as $x_{input} \in \mathbb{R}^{C \times L}$ and the target model output as $x_{pred} \in \mathbb{R}^{C \times L_p}$. The target output can represent various terms depending on the task, such as future sequences for the forecasting task or completed signals for the imputation task. The decomposition module processes the time series data into seasonal ($x_s$) and trend ($x_{trend}$) components. The *AdaWave* blocks then transform $x_s$ into a low-rank approximation $x_s^l$ and wavelet coefficients $c_l$ at different levels $l$. The channel-wise attention layer models the intermediate $x_s^l$ across channels to predict the targeted low-rank approximation $\hat{x}_s^l$. We reconstruct the predicted seasonal phase $\hat{x}_s$ from $c_l$ and $\hat{x}_s^l$ using inverse adaptive wavelet (*InvAdaWave*) blocks. The trend component $x_{trend}$ often exhibits alignment issues and discrepancies across variates, and we employ a grouped linear module that applies distinct linear heads to different channel groups, to enhance the quality of trend phase predictions. The network's final output is the sum of $\hat{x}_s$ and $\hat{x}_{trend}$.

*AdaWaveNet* offers several advantages, including multi-scale processing to mitigate non-stationary issues and a data-driven approach to learn wavelet coefficients through the lifting scheme adaptively. This adaptability is a key aspect of our proposed method. Additionally, the *AdaWave* and *InvAdaWave* blocks, based on convolutional layers, provide computational efficiency compared to the prior self-attention-based implementation of the wavelet transform (Zhou et al., 2022b). The grouped linear module further improves the modeling of the trend component, which tackles the discrepancies across channels or signal variates.

In the following subsections, we detail the proposed blocks, including the time series decomposition, *AdaWaveNet*, and Grouped Linear Module.

## 4.1 Time Series Decomposition

We use the additive time series decomposition method (Hamilton, 2020; Wu et al., 2021) to separate the time series sequences into their seasonal and trend components:

$$x_{input} = x_s + x_{trend} \tag{2}$$

In this equation, the trend component ($x_{trend}$) shows the overall direction and long-term movements of the data over time. It is often calculated using a moving average, which smooths out short-term fluctuations to highlight longer-term trends. The seasonal component ($x_s$) captures repeating patterns that occur within a specific period and helps identify systematic variations in the data.

To perform the decomposition, we first apply a moving average to the time series to estimate the trend component. Once the trend is identified, we subtract it from the original time series to isolate the seasonal component. By separating the data in this way, we can better understand and analyze the different scales of variation present in the time series. This decomposition is the first step in our multi-scale analysis framework.

## 4.2 Adaptive Wavelet Block

Given the seasonal component $x_s$ of a time series, we apply a Lifting Wavelet Transform (LWT) using Convolutional Neural Networks (CNNs) to refine features at various levels of granularity iteratively. These processes transform the seasonal component at level $l-1$ into a more refined level $l$, denoted as $x_s^l$, and generate the corresponding detail coefficients, $c^l$.

**Splitting Step**: The input $x_s^{l-1}$ (initially $x_s^0 = x_s$) is split into odd and even indexed components:

$$e^l = x_s^{l-1}[2i] \quad \forall i \in \mathbb{N} \tag{3}$$

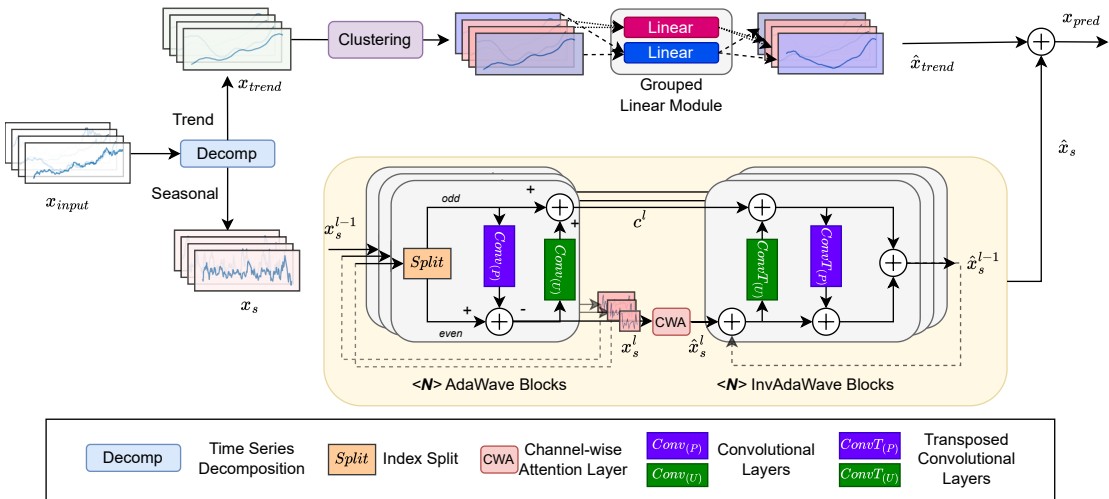

Figure 1: Illustration of the *AdaWaveNet* framework for time series analysis. The input sequence $x_{input}$ undergoes decomposition into trend ($x_{trend}$) and seasonal ($x_s$) components. The trend component is processed through a clustering algorithm followed by a grouped linear module to produce a refined trend prediction $\hat{x}_{trend}$. Concurrently, the seasonal component is processed through stacked *AdaWave* blocks, employing index splitting, convolutional layers, and a channel-wise attention layer to capture multi-scale features and generate a low-rank approximation $\hat{x}_s^l$. This is followed by inverse *AdaWave* blocks that reconstruct the seasonal prediction $\hat{x}_s$. The final predicted output $x_{pred}$ is obtained by summing the predicted seasonal and trend components.

$$o^l = x_s^{l-1}[2i + 1] \quad \forall i \in \mathbb{N} \tag{4}$$

**Convolutional Kernel-based Prediction and Update Steps**: Inspired by Huang & Fang (2021), we employ convolution operations as the wavelet filters in each split subset to extract approximations and coefficients. The learnable 1D convolution kernels are considered the ideal basis of the lifting scheme in our study. This operation can be represented as:

$$e'^l = \sigma(\mathbf{W}_e^l * e^l + b_e^l) \tag{5}$$

$$o'^l = \sigma(\mathbf{W}_o^l * o^l + b_o^l) \tag{6}$$

where $*$ denotes the convolution operation, $\mathbf{W}_e^l$ and $\mathbf{W}_o^l$ are the convolutional filter weights, and $b_e^l$ and $b_o^l$ are the biases for the even and odd components at level $l$, respectively.

In the prediction step, we use the even-indexed components to estimate the odd-indexed components. This is because, in many signals, there is a smooth transition between consecutive samples. The prediction step aims to estimate the finer details of the signal (odd indexed components) using a smoothed version (even indexed components) to capture the high-frequency variations by computing the detail coefficients $c^l$:

$$c^l = o^l - \sigma(\mathbf{W}_p^l * e^l + b_p^l) \tag{7}$$

Then, the update step utilizes these detail coefficients to refine the even indexed components:

$$e'^l = e^l + \sigma(\mathbf{W}_u^l * c^l + b_u^l) \tag{8}$$

These steps iteratively improve the signal representation in capturing the overall trend and the intricate details within the data.

### 4.3 Channel-wise Attention for Approximation Projection

To enhance the feature representation of the seasonal component $x_s^l$ specifically at the final level of the *AdaWave* blocks, a self-attention (SA) mechanism is employed, inspired by Liu et al. (2023a). The self-attention structure is applied to $x_s^N$, where $N$ denotes the final level of decomposition. This computation after the last layer of decomposition ensures efficient and focused refinement of the feature map of the low-rank approximation of the seasonal component, as the length of the sequences after $N$ blocks of *AdaWave* blocks becomes $(L/(2^N))$. The channel-wise attention mechanism operates on the channels of $x_s^N$ to refine its approximation, to project the processed seasonal component onto the targeted sequences as $\hat{x}_s^N$. Importantly, during this process, the detail coefficients ($c^N$) remain unchanged.

Focusing the channel-wise attention mechanism at the final decomposition layer is both computationally efficient and effective in capturing the essential characteristics of the time series. It allows the model to emphasize the global contextual information, which is crucial for the accurate representations of the seasonal components in complex time series data.

### 4.4 Inverse Adaptive Wavelet Blocks

To reconstruct the original seasonal component $\hat{x}_s$ from its refined representation $\hat{x}_s^l$ obtained after predicted approximation $\hat{x}_s^N$, we utilize an inverse process facilitated by Convolutional Transpose Networks. This inverse procedure employs transposed convolutional layers to upscale the feature maps and merge the detail coefficients with the upsampled seasonal components iteratively.

The inverse of the combining step involves an element-wise subtraction of the detail coefficients from the seasonal component at the current level $l$:

$$\hat{e}^l = \hat{x}_s^l - c^l \tag{9}$$

**Inverse Update and Prediction Steps**: The transposed convolution operations are applied to refine the split components and to estimate the original even indexed components:

$$e^l = \hat{e}^l - \sigma(\mathbf{W}_u^{l,T} * c^l + b_u^{l,T}) \tag{10}$$

The original odd indexed components are reconstructed by adding the predicted detail coefficients:

$$o^l = c^l + \sigma(\mathbf{W}_p^{l,T} * \tilde{e}^l + b_p^{l,T}) \tag{11}$$

where $\mathbf{W}_u^{l,T}$ and $\mathbf{W}_p^{l,T}$ denote the transposed convolutional filter weights, and $b_u^{l,T}$ and $b_p^{l,T}$ are the corresponding biases for the inverse update and prediction operations, respectively.

Finally, the odd and even indexed components are interleaved to reconstruct the seasonal component at the previous level $l-1$:

$$\hat{x}_s^{l-1}[2i] = e^l \quad \forall i \in \mathbb{N} \tag{12}$$

$$\hat{x}_s^{l-1}[2i+1] = \hat{o}^l \quad \forall i \in \mathbb{N} \tag{13}$$

### 4.5 Grouped Linear Module

We process the trend component $x_{trend}$ using a two-step approach: clustering the channels and applying distinct linear projections based on the clustering labels. This method effectively captures and models the long-term progression of the trend component.

In the first step, we group the channels of $x_{trend}$ using a clustering algorithm, specifically K-means. This clustering step helps to identify patterns and group similar channels together. For example, in the diagram as in Figure 1, channels are clustered into different groups according to their characteristics. This step is crucial for understanding the temporal patterns within the trend data that might indicate different states across variates.

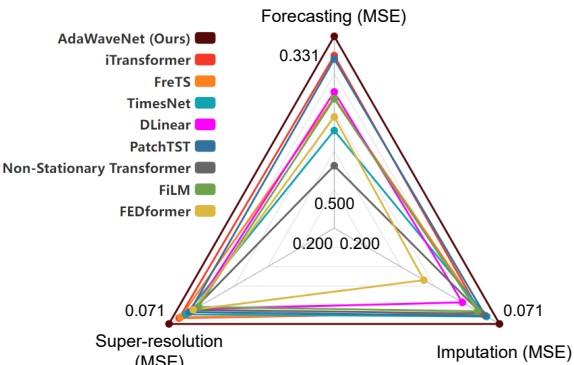

Figure 2: Comparison of model performances in forecasting, imputation, and super-resolution.

Following the clustering, each group of channels is connected with a linear projection layer. This step ensures that the model can apply specific linear projections to different clusters of the trend data. This design aims to help the model accurately capture different patterns within the trend data.

This dual approach of clustering followed by grouped linear transformations is particularly effective in dealing with complex time series data. By initially clustering the trend components into different groups, the model can recognize and adapt to different underlying patterns. Subsequent application of distinct linear projections to each cluster further enhances the model's ability to represent and forecast the trend component accurately.

## 5  Experiments

We conduct experiments on different time series analysis tasks to evaluate the proposed *AdaWaveNet*, including forecasting, imputation, and the newly proposed super-resolution tasks.

*AdaWaveNet* is extensively benchmarked against established models from recent literature. For models related to wavelet and frequency domain enhancements, comparisons include FreTS (Yi et al., 2023), FiLM (Zhou et al., 2022a), TimesNet (Wu et al., 2022), and FEDformer (Zhou et al., 2022b). Additionally, models previously recognized for state-of-the-art (SOTA) performances, such as iTransformer (Liu et al., 2023a), DLinear (Zeng et al., 2023), and PatchTST (Nie et al., 2022), are included as experimental baselines. The Non-stationary Transformer (Stationary) (Liu et al., 2022b), known for addressing non-stationary issues in time series, is also featured for comparison.

Figure 2 presents the aggregated results across the forecasting, imputation, and super-resolution tasks. The results indicate that our proposed *AdaWaveNet* method achieves SOTA performance in all three areas of time series analysis.

### 5.1  Time Series Forecasting

Forecasting is essential in time series applications, such as weather, traffic, exchange rate, etc. In this section, we conduct extensive experiments to evaluate the forecasting performance of the proposed *AdaWaveNet* on varying-domain datasets. Following prior studies such as iTransformer (Liu et al., 2023a) and TimesNet (Wu et al., 2022), we adopt a long-term forecasting setting with datasets including traffic, electricity (ECL), exchange rate, weather, solar energy, and electricity transformer temperature (ETT). For each dataset, we set the predicting length $L_p$ with {96, 192, 336, 720} with the inputting observation length equal to the predicting length.

The performances of evaluations are shown in Table 1. The proposed *AdaWaveNet* achieves promising performances in both MSE and MAE across various datasets. In particular, compared to the prior frequency domain-enhanced methods, *AdaWaveNet* outperforms 7.7% in MSE and 6.2% in MAE when compared to the best performances in all these methods. For the SOTA methods such as iTransformer (Liu et al., 2023a) and

Table 1: Forecasting task. The prediction lengths for all datasets are established at {96, 192, 336, 720}, with the past sequence length matching the prediction lengths. Evaluation metrics include Mean Squared Error (MSE) and Mean Absolute Error (MAE). The lowest MSE is indicated in bold red, while the second lowest is underlined in blue.

| Model (Year) | | AdaWaveNet (Ours) | | iTransformer (2023a) | | FreTS (2023) | | TimesNet (2022) | | DLinear (2023) | | PatchTST (2022) | | Stationary (2022b) | | FiLM (2022a) | | FEDformer (2022b) | |
|---|---|---|---|---|---|---|---|---|---|---|---|---|---|---|---|---|---|---|---|
| Pred Length | | MSE | MAE | MSE | MAE | MSE | MAE | MSE | MAE | MSE | MAE | MSE | MAE | MSE | MAE | MSE | MAE | MSE | MAE |
| ECL | 96 | 0.146 | 0.248 | 0.151 | 0.243 | 0.189 | 0.277 | 0.165 | 0.269 | 0.210 | 0.302 | 0.181 | 0.268 | 0.167 | 0.269 | 0.398 | 0.452 | 0.193 | 0.308 |
| | 192 | 0.158 | 0.260 | 0.156 | 0.258 | 0.164 | 0.261 | 0.185 | 0.289 | 0.174 | 0.275 | 0.158 | 0.254 | 0.193 | 0.295 | 0.266 | 0.361 | 0.208 | 0.326 |
| | 336 | 0.171 | 0.268 | 0.177 | 0.274 | 0.202 | 0.289 | 0.196 | 0.301 | 0.176 | 0.278 | 0.176 | 0.278 | 0.207 | 0.310 | 0.281 | 0.377 | 0.224 | 0.339 |
| | 720 | 0.196 | 0.292 | 0.201 | 0.299 | 0.225 | 0.322 | 0.215 | 0.317 | 0.201 | 0.300 | 0.207 | 0.310 | 0.214 | 0.317 | 0.289 | 0.385 | 0.242 | 0.357 |
| | Avg. | 0.168 | 0.267 | 0.171 | 0.269 | 0.195 | 0.287 | 0.190 | 0.294 | 0.190 | 0.289 | 0.181 | 0.278 | 0.195 | 0.298 | 0.309 | 0.394 | 0.217 | 0.333 |
| Weather | 96 | 0.169 | 0.215 | 0.177 | 0.217 | 0.173 | 0.231 | 0.172 | 0.220 | 0.196 | 0.254 | 0.176 | 0.217 | 0.197 | 0.238 | 0.193 | 0.234 | 0.219 | 0.298 |
| | 192 | 0.203 | 0.245 | 0.213 | 0.253 | 0.204 | 0.268 | 0.222 | 0.266 | 0.227 | 0.286 | 0.212 | 0.258 | 0.264 | 0.298 | 0.230 | 0.266 | 0.265 | 0.341 |
| | 336 | 0.248 | 0.286 | 0.256 | 0.289 | 0.255 | 0.313 | 0.286 | 0.314 | 0.262 | 0.313 | 0.256 | 0.291 | 0.316 | 0.330 | 0.266 | 0.295 | 0.316 | 0.380 |
| | 720 | 0.313 | 0.336 | 0.325 | 0.343 | 0.320 | 0.356 | 0.375 | 0.373 | 0.318 | 0.359 | 0.321 | 0.339 | 0.400 | 0.378 | 0.322 | 0.338 | 0.372 | 0.403 |
| | Avg. | 0.233 | 0.271 | 0.243 | 0.276 | 0.238 | 0.292 | 0.264 | 0.293 | 0.251 | 0.303 | 0.241 | 0.276 | 0.294 | 0.311 | 0.253 | 0.283 | 0.293 | 0.356 |
| Traffic | 96 | 0.417 | 0.291 | 0.396 | 0.271 | 0.525 | 0.333 | 0.592 | 0.319 | 0.652 | 0.397 | 0.544 | 0.359 | 0.610 | 0.341 | 0.647 | 0.384 | 0.585 | 0.363 |
| | 192 | 0.401 | 0.281 | 0.400 | 0.280 | 0.514 | 0.329 | 0.592 | 0.321 | 0.622 | 0.354 | 0.547 | 0.361 | 0.621 | 0.348 | 0.462 | 0.302 | 0.586 | 0.366 |
| | 336 | 0.407 | 0.284 | 0.415 | 0.286 | 0.527 | 0.341 | 0.611 | 0.330 | 0.624 | 0.355 | 0.555 | 0.368 | 0.628 | 0.344 | 0.447 | 0.305 | 0.597 | 0.369 |
| | 720 | 0.433 | 0.297 | 0.428 | 0.301 | 0.546 | 0.359 | 0.626 | 0.344 | 0.664 | 0.408 | 0.603 | 0.393 | 0.647 | 0.384 | 0.485 | 0.321 | 0.615 | 0.390 |
| | Avg. | 0.415 | 0.288 | 0.410 | 0.285 | 0.528 | 0.341 | 0.605 | 0.329 | 0.641 | 0.379 | 0.562 | 0.370 | 0.627 | 0.354 | 0.510 | 0.328 | 0.596 | 0.372 |
| Exchange | 96 | 0.086 | 0.204 | 0.087 | 0.209 | 0.088 | 0.214 | 0.113 | 0.243 | 0.093 | 0.227 | 0.093 | 0.212 | 0.150 | 0.265 | 0.166 | 0.307 | 0.147 | 0.276 |
| | 192 | 0.188 | 0.310 | 0.197 | 0.320 | 0.211 | 0.347 | 0.243 | 0.361 | 0.182 | 0.323 | 0.188 | 0.312 | 0.261 | 0.375 | 0.224 | 0.345 | 0.260 | 0.387 |
| | 336 | 0.360 | 0.437 | 0.398 | 0.452 | 0.572 | 0.576 | 0.466 | 0.502 | 0.391 | 0.477 | 0.328 | 0.415 | 0.633 | 0.581 | 0.400 | 0.467 | 0.502 | 0.544 |
| | 720 | 1.288 | 0.862 | 1.370 | 0.849 | 1.576 | 0.972 | 1.796 | 1.044 | 1.364 | 0.888 | 1.144 | 0.800 | 1.357 | 0.875 | 1.200 | 0.883 | 1.491 | 0.954 |
| | Avg. | 0.481 | 0.453 | 0.513 | 0.458 | 0.612 | 0.527 | 0.655 | 0.543 | 0.508 | 0.479 | 0.438 | 0.435 | 0.600 | 0.524 | 0.498 | 0.501 | 0.600 | 0.540 |
| Solar | 96 | 0.199 | 0.254 | 0.211 | 0.256 | 0.227 | 0.292 | 0.249 | 0.290 | 0.286 | 0.374 | 0.223 | 0.271 | 0.216 | 0.251 | 0.309 | 0.334 | 0.243 | 0.343 |
| | 192 | 0.207 | 0.262 | 0.216 | 0.269 | 0.213 | 0.279 | 0.238 | 0.281 | 0.261 | 0.330 | 0.211 | 0.258 | 0.219 | 0.252 | 0.275 | 0.282 | 0.244 | 0.346 |
| | 336 | 0.216 | 0.269 | 0.220 | 0.272 | 0.242 | 0.297 | 0.242 | 0.285 | 0.270 | 0.325 | 0.214 | 0.273 | 0.229 | 0.272 | 0.288 | 0.294 | 0.251 | 0.352 |
| | 720 | 0.214 | 0.263 | 0.223 | 0.286 | 0.255 | 0.306 | 0.245 | 0.287 | 0.237 | 0.296 | 0.221 | 0.282 | 0.227 | 0.275 | 0.291 | 0.298 | 0.252 | 0.355 |
| | Avg. | 0.209 | 0.262 | 0.218 | 0.271 | 0.234 | 0.294 | 0.244 | 0.286 | 0.264 | 0.331 | 0.217 | 0.271 | 0.223 | 0.263 | 0.291 | 0.302 | 0.248 | 0.349 |
| ETTh1 | 96 | 0.384 | 0.396 | 0.420 | 0.428 | 0.412 | 0.430 | 0.400 | 0.420 | 0.385 | 0.431 | 0.384 | 0.402 | 0.559 | 0.505 | 0.387 | 0.399 | 0.377 | 0.418 |
| | 192 | 0.437 | 0.431 | 0.463 | 0.456 | 0.467 | 0.461 | 0.564 | 0.526 | 0.430 | 0.443 | 0.426 | 0.428 | 0.698 | 0.575 | 0.437 | 0.430 | 0.422 | 0.451 |
| | 336 | 0.445 | 0.441 | 0.489 | 0.475 | 0.501 | 0.493 | 0.509 | 0.495 | 0.437 | 0.453 | 0.460 | 0.456 | 0.664 | 0.568 | 0.459 | 0.455 | 0.451 | 0.453 |
| | 720 | 0.510 | 0.497 | 0.600 | 0.565 | 0.602 | 0.573 | 0.708 | 0.615 | 0.492 | 0.510 | 0.505 | 0.470 | 0.713 | 0.615 | 0.509 | 0.501 | 0.497 | 0.499 |
| | Avg. | 0.444 | 0.441 | 0.493 | 0.481 | 0.496 | 0.489 | 0.545 | 0.514 | 0.436 | 0.459 | 0.444 | 0.439 | 0.659 | 0.566 | 0.448 | 0.446 | 0.437 | 0.455 |
| ETTm1 | 96 | 0.326 | 0.366 | 0.354 | 0.381 | 0.343 | 0.376 | 0.330 | 0.371 | 0.345 | 0.371 | 0.334 | 0.368 | 0.422 | 0.415 | 0.351 | 0.370 | 0.379 | 0.419 |
| | 192 | 0.335 | 0.370 | 0.355 | 0.384 | 0.364 | 0.394 | 0.396 | 0.406 | 0.342 | 0.368 | 0.339 | 0.368 | 0.466 | 0.446 | 0.388 | 0.404 | 0.415 | 0.428 |
| | 336 | 0.375 | 0.394 | 0.384 | 0.406 | 0.393 | 0.409 | 0.403 | 0.422 | 0.370 | 0.386 | 0.367 | 0.392 | 0.555 | 0.496 | 0.400 | 0.420 | 0.417 | 0.431 |
| | 720 | 0.439 | 0.428 | 0.448 | 0.449 | 0.466 | 0.457 | 0.456 | 0.455 | 0.420 | 0.422 | 0.456 | 0.447 | 0.615 | 0.531 | 0.447 | 0.439 | 0.484 | 0.479 |
| | Avg. | 0.369 | 0.390 | 0.385 | 0.405 | 0.392 | 0.409 | 0.396 | 0.414 | 0.369 | 0.387 | 0.374 | 0.394 | 0.515 | 0.472 | 0.397 | 0.408 | 0.424 | 0.439 |
| 1st Count | | 18 | 21 | 5 | 3 | 0 | 0 | 0 | 0 | 5 | 3 | 5 | 5 | 0 | 2 | 0 | 0 | 2 | 1 |

PatchTST (Nie et al., 2022), the evaluation results also suggest that our proposed method outperforms them in the forecasting task.

## 5.2 Time Series Imputation

During time series data collection, various factors can disrupt continuous observation and monitoring and lead to missing values within the dataset. For example, factors such as the malfunction of the devices and interference of signals can all trigger data quality issues, e.g., missing observation. Therefore, in this study, we also investigate the capability of the proposed method in time series imputation tasks. Following the evaluation strategy as Wu et al. (2022), we conduct extensive experiments on the datasets with controlled masking rates under a random missing setting. We further extend the experimental strategy by introducing the extended missingness. In (Wu et al., 2022), the missingness was crafted by randomly masking timestamps with a certain ratio that simulates sporadic data loss; whereas the extended missingness masks contiguous subsequences of the original signals across all channels, which emulates the prolonged periods of missing data. Refer to Appendix A.2 for the detailed descriptions of two masking methods. In this experiment, we examine the proposed method and the baselines on the weather and electricity data. Also, considering the common missingness of sensing data of the wearable sensors Xu et al. (2022), the experiments cover biobehavioral datasets, including 12-lead electrocardiogram (ECG) from the PTB-XL dataset Wagner et al. (2020), and electroencephalogram (EEG) from the Sleep-EDFE dataset Kemp et al. (2000).

Table 2 shows the imputation performance for each dataset. The observed results show the two different masking strategies create different levels of challenges for the imputation tasks. The extended masking task introduces larger errors for all the models. For both imputations, our proposed method shows competitive performances on the PTB-XL and Sleep-EDFE datasets, where the sequence lengths are significantly longer than the ECL and weather datasets. TimesNet (Wu et al., 2022) outperforms all the other models in ECL and Weather random imputation tasks; whereas *AdaWaveNet* shows substantial improvement compared to the other baselines on ECL and Weather data for the random masked imputation task. Visualization of imputed examples can be found in Appendix C.2.

Table 2: Imputation task. Experiments are conducted on two types of imputation - random and extended. In each case, we mask {12.5%, 25%, 37.5%, 50%} of time points or segments randomly from the original sequences. For the ECL and Weather datasets, sequence lengths are set to 96, while for the PTB-XL and Sleep-EDFE datasets, the lengths are 1000 and 3000, respectively. Evaluation metrics include Mean Squared Error (MSE) and Mean Absolute Error (MAE). The lowest MSE is marked in bold red, and the second lowest is underlined in blue.

| | | Model (Year) | AdaWaveNet (Ours) | | iTransformer (2023a) | | FreTS (2023) | | TimesNet (2022) | | DLinear (2023) | | PatchTST (2022) | | Stationary (2022b) | | FiLM (2022a) | | FEDformer (2022b) | |
|---|---|---|---|---|---|---|---|---|---|---|---|---|---|---|---|---|---|---|---|---|
| | | Mask Ratio | MSE | MAE | MSE | MAE | MSE | MAE | MSE | MAE | MSE | MAE | MSE | MAE | MSE | MAE | MSE | MAE | MSE | MAE |
| Random | ECL | 0.125 | 0.080 | 0.202 | 0.089 | 0.210 | 0.102 | 0.218 | 0.089 | 0.205 | 0.123 | 0.251 | 0.077 | 0.194 | 0.093 | 0.210 | 0.095 | 0.216 | 0.107 | 0.237 |
| | | 0.25 | 0.091 | 0.205 | 0.101 | 0.229 | 0.107 | 0.225 | 0.092 | 0.208 | 0.114 | 0.424 | 0.093 | 0.215 | 0.097 | 0.214 | 0.102 | 0.246 | 0.120 | 0.251 |
| | | 0.375 | 0.108 | 0.219 | 0.124 | 0.256 | 0.128 | 0.243 | 0.096 | 0.213 | 0.141 | 0.273 | 0.110 | 0.236 | 0.102 | 0.220 | 0.118 | 0.241 | 0.136 | 0.266 |
| | | 0.5 | 0.122 | 0.234 | 0.152 | 0.286 | 0.147 | 0.271 | 0.102 | 0.221 | 0.173 | 0.303 | 0.114 | 0.240 | 0.108 | 0.228 | 0.135 | 0.258 | 0.158 | 0.284 |
| | | Avg. | 0.100 | 0.215 | 0.117 | 0.245 | 0.121 | 0.239 | 0.095 | 0.212 | 0.138 | 0.313 | 0.099 | 0.221 | 0.100 | 0.218 | 0.112 | 0.240 | 0.130 | 0.260 |
| | Weather | 0.125 | 0.033 | 0.059 | 0.029 | 0.062 | 0.023 | 0.049 | 0.025 | 0.045 | 0.039 | 0.084 | 0.028 | 0.065 | 0.027 | 0.051 | 0.031 | 0.066 | 0.044 | 0.110 |
| | | 0.25 | 0.041 | 0.077 | 0.046 | 0.083 | 0.044 | 0.080 | 0.029 | 0.052 | 0.048 | 0.103 | 0.040 | 0.079 | 0.029 | 0.056 | 0.042 | 0.086 | 0.062 | 0.160 |
| | | 0.375 | 0.050 | 0.099 | 0.055 | 0.105 | 0.059 | 0.120 | 0.031 | 0.057 | 0.057 | 0.117 | 0.051 | 0.096 | 0.033 | 0.062 | 0.055 | 0.111 | 0.107 | 0.231 |
| | | 0.5 | 0.062 | 0.129 | 0.068 | 0.136 | 0.072 | 0.142 | 0.034 | 0.062 | 0.066 | 0.134 | 0.058 | 0.107 | 0.037 | 0.068 | 0.073 | 0.136 | 0.183 | 0.311 |
| | | Avg. | 0.047 | 0.091 | 0.050 | 0.097 | 0.050 | 0.098 | 0.030 | 0.054 | 0.053 | 0.110 | 0.044 | 0.087 | 0.032 | 0.059 | 0.050 | 0.100 | 0.099 | 0.203 |
| | PTB-XL | 0.125 | 0.017 | 0.028 | 0.032 | 0.044 | 0.029 | 0.040 | 0.025 | 0.031 | 0.042 | 0.049 | 0.024 | 0.035 | 0.022 | 0.033 | 0.028 | 0.039 | 0.035 | 0.051 |
| | | 0.25 | 0.024 | 0.037 | 0.044 | 0.055 | 0.040 | 0.052 | 0.030 | 0.044 | 0.058 | 0.071 | 0.035 | 0.046 | 0.033 | 0.045 | 0.040 | 0.052 | 0.052 | 0.064 |
| | | 0.375 | 0.029 | 0.039 | 0.060 | 0.069 | 0.052 | 0.065 | 0.034 | 0.047 | 0.057 | 0.068 | 0.044 | 0.058 | 0.045 | 0.059 | 0.049 | 0.062 | 0.074 | 0.089 |
| | | 0.5 | 0.044 | 0.058 | 0.077 | 0.085 | 0.063 | 0.077 | 0.041 | 0.056 | 0.073 | 0.087 | 0.057 | 0.069 | 0.057 | 0.066 | 0.063 | 0.075 | 0.091 | 0.103 |
| | | Avg. | 0.029 | 0.041 | 0.053 | 0.063 | 0.046 | 0.059 | 0.033 | 0.045 | 0.058 | 0.069 | 0.040 | 0.052 | 0.039 | 0.051 | 0.045 | 0.057 | 0.063 | 0.077 |
| | Sleep-EDFE | 0.125 | 0.024 | 0.036 | 0.033 | 0.047 | 0.027 | 0.039 | 0.041 | 0.055 | 0.036 | 0.050 | 0.031 | 0.038 | 0.047 | 0.065 | 0.036 | 0.050 | 0.052 | 0.068 |
| | | 0.25 | 0.031 | 0.040 | 0.042 | 0.058 | 0.037 | 0.051 | 0.046 | 0.063 | 0.045 | 0.058 | 0.040 | 0.049 | 0.059 | 0.072 | 0.044 | 0.057 | 0.048 | 0.066 |
| | | 0.375 | 0.037 | 0.048 | 0.052 | 0.062 | 0.048 | 0.062 | 0.050 | 0.067 | 0.056 | 0.071 | 0.051 | 0.062 | 0.070 | 0.084 | 0.054 | 0.068 | 0.067 | 0.084 |
| | | 0.5 | 0.043 | 0.055 | 0.061 | 0.070 | 0.059 | 0.073 | 0.052 | 0.074 | 0.068 | 0.085 | 0.057 | 0.069 | 0.081 | 0.095 | 0.063 | 0.078 | 0.082 | 0.102 |
| | | Avg. | 0.034 | 0.045 | 0.047 | 0.059 | 0.043 | 0.056 | 0.047 | 0.065 | 0.051 | 0.066 | 0.045 | 0.055 | 0.064 | 0.079 | 0.049 | 0.063 | 0.062 | 0.080 |
| Extended | ECL | 0.125 | 0.098 | 0.207 | 0.109 | 0.217 | 0.130 | 0.217 | 0.114 | 0.227 | 0.168 | 0.276 | 0.120 | 0.229 | 0.115 | 0.226 | 0.126 | 0.219 | 0.152 | 0.254 |
| | | 0.25 | 0.104 | 0.207 | 0.116 | 0.218 | 0.128 | 0.222 | 0.117 | 0.225 | 0.138 | 0.230 | 0.131 | 0.231 | 0.124 | 0.233 | 0.127 | 0.228 | 0.155 | 0.261 |
| | | 0.375 | 0.121 | 0.228 | 0.138 | 0.241 | 0.160 | 0.257 | 0.139 | 0.243 | 0.194 | 0.292 | 0.163 | 0.263 | 0.134 | 0.243 | 0.151 | 0.255 | 0.161 | 0.270 |
| | | 0.5 | 0.126 | 0.230 | 0.143 | 0.241 | 0.165 | 0.262 | 0.140 | 0.246 | 0.186 | 0.266 | 0.172 | 0.263 | 0.137 | 0.245 | 0.156 | 0.256 | 0.177 | 0.293 |
| | | Avg. | 0.112 | 0.218 | 0.127 | 0.229 | 0.146 | 0.240 | 0.128 | 0.235 | 0.172 | 0.266 | 0.147 | 0.247 | 0.128 | 0.237 | 0.140 | 0.239 | 0.161 | 0.270 |
| | Weather | 0.125 | 0.067 | 0.091 | 0.074 | 0.113 | 0.072 | 0.112 | 0.216 | 0.257 | 0.072 | 0.106 | 0.082 | 0.108 | 0.086 | 0.122 | 0.101 | 0.138 | 0.138 | 0.197 |
| | | 0.25 | 0.084 | 0.127 | 0.082 | 0.118 | 0.090 | 0.135 | 0.086 | 0.118 | 0.097 | 0.149 | 0.079 | 0.118 | 0.101 | 0.158 | 0.097 | 0.143 | 0.154 | 0.218 |
| | | 0.375 | 0.098 | 0.154 | 0.102 | 0.139 | 0.115 | 0.171 | 0.101 | 0.142 | 0.121 | 0.184 | 0.096 | 0.138 | 0.113 | 0.176 | 0.115 | 0.167 | 0.177 | 0.229 |
| | | 0.5 | 0.107 | 0.170 | 0.119 | 0.159 | 0.127 | 0.182 | 0.113 | 0.157 | 0.135 | 0.202 | 0.112 | 0.153 | 0.125 | 0.191 | 0.128 | 0.181 | 0.186 | 0.231 |
| | | Avg. | 0.089 | 0.136 | 0.094 | 0.132 | 0.101 | 0.150 | 0.129 | 0.169 | 0.106 | 0.160 | 0.092 | 0.129 | 0.106 | 0.162 | 0.110 | 0.157 | 0.164 | 0.219 |
| | PTB-XL | 0.125 | 0.049 | 0.062 | 0.069 | 0.087 | 0.055 | 0.071 | 0.044 | 0.069 | 0.075 | 0.107 | 0.064 | 0.083 | 0.062 | 0.083 | 0.061 | 0.082 | 0.071 | 0.096 |
| | | 0.25 | 0.063 | 0.080 | 0.077 | 0.098 | 0.071 | 0.092 | 0.065 | 0.090 | 0.082 | 0.118 | 0.075 | 0.097 | 0.071 | 0.092 | 0.074 | 0.097 | 0.089 | 0.106 |
| | | 0.375 | 0.074 | 0.089 | 0.094 | 0.106 | 0.085 | 0.124 | 0.089 | 0.108 | 0.103 | 0.125 | 0.096 | 0.111 | 0.083 | 0.106 | 0.092 | 0.110 | 0.109 | 0.133 |
| | | 0.5 | 0.089 | 0.122 | 0.112 | 0.135 | 0.099 | 0.136 | 0.104 | 0.134 | 0.114 | 0.141 | 0.110 | 0.126 | 0.097 | 0.114 | 0.105 | 0.132 | 0.118 | 0.147 |
| | | Avg. | 0.069 | 0.088 | 0.088 | 0.107 | 0.078 | 0.106 | 0.076 | 0.100 | 0.094 | 0.123 | 0.086 | 0.104 | 0.078 | 0.099 | 0.083 | 0.105 | 0.097 | 0.121 |
| | Sleep-EDFE | 0.125 | 0.066 | 0.087 | 0.092 | 0.112 | 0.070 | 0.098 | 0.083 | 0.104 | 0.102 | 0.134 | 0.079 | 0.096 | 0.113 | 0.149 | 0.089 | 0.115 | 0.103 | 0.140 |
| | | 0.25 | 0.082 | 0.105 | 0.114 | 0.146 | 0.085 | 0.110 | 0.106 | 0.133 | 0.117 | 0.158 | 0.096 | 0.120 | 0.125 | 0.164 | 0.105 | 0.136 | 0.116 | 0.155 |
| | | 0.375 | 0.104 | 0.137 | 0.119 | 0.154 | 0.110 | 0.142 | 0.124 | 0.155 | 0.139 | 0.188 | 0.104 | 0.132 | 0.133 | 0.177 | 0.122 | 0.160 | 0.142 | 0.197 |
| | | 0.5 | 0.109 | 0.141 | 0.131 | 0.188 | 0.121 | 0.157 | 0.142 | 0.191 | 0.155 | 0.207 | 0.127 | 0.163 | 0.147 | 0.196 | 0.134 | 0.180 | 0.139 | 0.195 |
| | | Avg. | 0.090 | 0.118 | 0.114 | 0.150 | 0.097 | 0.127 | 0.114 | 0.146 | 0.128 | 0.172 | 0.102 | 0.128 | 0.130 | 0.172 | 0.112 | 0.148 | 0.125 | 0.172 |

Table 3: Super-resolution task. Super-resolution upsampling ratios are set at {2, 5, 10}. Experimentally, sequence lengths are fixed at 200 for ETTm1, ETTh1, and Traffic datasets, and at 1000, 3000, and 960 for PTB-XL, Sleep-EDFE, and CLAS datasets, respectively. Evaluation metrics include Mean Squared Error (MSE) and Mean Absolute Error (MAE), with all results being averages over 4 masking ratios. The lowest MSE is highlighted in bold red, while the second lowest is underlined in blue.

| Model (Year) | | AdaWaveNet (Ours) | | iTransformer (2023a) | | FreTS (2023) | | TimesNet (2022) | | DLinear (2023) | | PatchTST (2022) | | Stationary (2022b) | | FiLM (2022a) | | FEDformer (2022b) | |
|---|---|---|---|---|---|---|---|---|---|---|---|---|---|---|---|---|---|---|---|---|
| SR Ratio | | MSE | MAE | MSE | MAE | MSE | MAE | MSE | MAE | MSE | MAE | MSE | MAE | MSE | MAE | MSE | MAE | MSE | MAE |
| ETTm1 | 2 | 0.016 | 0.085 | 0.021 | 0.097 | 0.024 | 0.102 | 0.027 | 0.096 | 0.044 | 0.123 | 0.035 | 0.111 | 0.031 | 0.104 | 0.051 | 0.136 | 0.037 | 0.109 |
| | 5 | 0.035 | 0.101 | 0.036 | 0.110 | 0.040 | 0.115 | 0.039 | 0.112 | 0.070 | 0.160 | 0.057 | 0.146 | 0.038 | 0.117 | 0.047 | 0.121 | 0.055 | 0.158 |
| | 10 | 0.058 | 0.151 | 0.065 | 0.173 | 0.077 | 0.179 | 0.054 | 0.147 | 0.087 | 0.194 | 0.077 | 0.182 | 0.062 | 0.166 | 0.070 | 0.172 | 0.081 | 0.190 |
| | Avg. | 0.036 | 0.112 | 0.041 | 0.127 | 0.047 | 0.132 | 0.040 | 0.118 | 0.067 | 0.159 | 0.056 | 0.146 | 0.044 | 0.129 | 0.056 | 0.143 | 0.058 | 0.152 |
| ETTh1 | 2 | 0.039 | 0.112 | 0.046 | 0.125 | 0.043 | 0.115 | 0.051 | 0.123 | 0.063 | 0.134 | 0.054 | 0.128 | 0.051 | 0.130 | 0.048 | 0.129 | 0.037 | 0.109 |
| | 5 | 0.093 | 0.193 | 0.107 | 0.210 | 0.099 | 0.200 | 0.102 | 0.205 | 0.128 | 0.231 | 0.110 | 0.217 | 0.090 | 0.194 | 0.100 | 0.203 | 0.107 | 0.208 |
| | 10 | 0.178 | 0.270 | 0.202 | 0.291 | 0.190 | 0.287 | 0.187 | 0.287 | 0.202 | 0.304 | 0.196 | 0.292 | 0.168 | 0.266 | 0.189 | 0.294 | 0.201 | 0.297 |
| | Avg. | 0.103 | 0.192 | 0.118 | 0.209 | 0.111 | 0.201 | 0.113 | 0.205 | 0.131 | 0.223 | 0.120 | 0.212 | 0.103 | 0.197 | 0.112 | 0.209 | 0.115 | 0.205 |
| Traffic | 2 | 0.107 | 0.096 | 0.114 | 0.104 | 0.127 | 0.119 | 0.133 | 0.125 | 0.141 | 0.136 | 0.124 | 0.111 | 0.157 | 0.155 | 0.162 | 0.154 | 0.155 | 0.142 |
| | 5 | 0.210 | 0.197 | 0.208 | 0.193 | 0.226 | 0.213 | 0.241 | 0.221 | 0.239 | 0.222 | 0.219 | 0.197 | 0.264 | 0.250 | 0.277 | 0.258 | 0.256 | 0.233 |
| | 10 | 0.322 | 0.281 | 0.329 | 0.285 | 0.351 | 0.307 | 0.369 | 0.322 | 0.355 | 0.319 | 0.337 | 0.295 | 0.383 | 0.346 | 0.401 | 0.377 | 0.363 | 0.336 |
| | Avg. | 0.213 | 0.191 | 0.217 | 0.194 | 0.235 | 0.213 | 0.248 | 0.223 | 0.245 | 0.226 | 0.227 | 0.201 | 0.268 | 0.250 | 0.280 | 0.263 | 0.258 | 0.237 |
| PTB-XL | 2 | 0.007 | 0.016 | 0.009 | 0.019 | 0.007 | 0.015 | 0.006 | 0.015 | 0.011 | 0.021 | 0.014 | 0.020 | 0.012 | 0.025 | 0.008 | 0.018 | 0.010 | 0.022 |
| | 5 | 0.016 | 0.023 | 0.019 | 0.028 | 0.017 | 0.023 | 0.015 | 0.021 | 0.023 | 0.037 | 0.026 | 0.036 | 0.023 | 0.037 | 0.019 | 0.027 | 0.020 | 0.029 |
| | 10 | 0.033 | 0.045 | 0.035 | 0.046 | 0.036 | 0.048 | 0.030 | 0.042 | 0.045 | 0.062 | 0.052 | 0.069 | 0.042 | 0.061 | 0.039 | 0.052 | 0.037 | 0.050 |
| | Avg. | 0.019 | 0.028 | 0.021 | 0.031 | 0.020 | 0.029 | 0.017 | 0.026 | 0.026 | 0.040 | 0.031 | 0.042 | 0.026 | 0.041 | 0.022 | 0.032 | 0.022 | 0.034 |
| Sleep-EDFE | 2 | 0.011 | 0.133 | 0.017 | 0.140 | 0.015 | 0.137 | 0.027 | 0.160 | 0.022 | 0.155 | 0.025 | 0.156 | 0.031 | 0.166 | 0.020 | 0.150 | 0.027 | 0.158 |
| | 5 | 0.022 | 0.141 | 0.036 | 0.168 | 0.028 | 0.154 | 0.035 | 0.179 | 0.035 | 0.177 | 0.038 | 0.182 | 0.045 | 0.184 | 0.039 | 0.173 | 0.042 | 0.179 |
| | 10 | 0.027 | 0.159 | 0.041 | 0.177 | 0.036 | 0.170 | 0.044 | 0.181 | 0.049 | 0.190 | 0.048 | 0.194 | 0.064 | 0.212 | 0.053 | 0.202 | 0.061 | 0.222 |
| | Avg. | 0.020 | 0.144 | 0.031 | 0.162 | 0.026 | 0.154 | 0.035 | 0.173 | 0.035 | 0.174 | 0.037 | 0.177 | 0.047 | 0.187 | 0.037 | 0.175 | 0.043 | 0.186 |
| CLAS | 2 | 0.018 | 0.048 | 0.025 | 0.057 | 0.023 | 0.055 | 0.037 | 0.083 | 0.022 | 0.054 | 0.033 | 0.078 | 0.044 | 0.081 | 0.030 | 0.071 | 0.029 | 0.062 |
| | 5 | 0.034 | 0.079 | 0.047 | 0.084 | 0.041 | 0.082 | 0.046 | 0.099 | 0.039 | 0.088 | 0.054 | 0.098 | 0.046 | 0.082 | 0.052 | 0.103 | 0.039 | 0.084 |
| | 10 | 0.051 | 0.101 | 0.066 | 0.120 | 0.062 | 0.111 | 0.060 | 0.107 | 0.056 | 0.114 | 0.072 | 0.135 | 0.066 | 0.117 | 0.078 | 0.144 | 0.062 | 0.116 |
| | Avg. | 0.034 | 0.076 | 0.046 | 0.087 | 0.042 | 0.083 | 0.048 | 0.096 | 0.039 | 0.085 | 0.053 | 0.104 | 0.052 | 0.093 | 0.053 | 0.106 | 0.043 | 0.087 |

Table 4: The averaged results of model ablation with mean squared error as the evaluation metric. "GL" denotes the Grouped Linear module, "CWA" indicates Channel-wise Attention, and "AWB" signifies the *AdaWave* component. Tasks are marked as (F) for forecasting and (I) for extended imputation. The highest MSE is highlighted in bold red, while the second highest is underlined in blue. Refer to Table 8 in the Appendix for comprehensive results.

| w/o | - | GL | RevIN | CWA | AWB |
|---|---|---|---|---|---|
| Weather (F) | 0.233 | 0.240 | 0.246 | **0.256** | 0.255 |
| Traffic (F) | 0.415 | 0.425 | 0.422 | **0.519** | 0.473 |
| ECL (I) | 0.112 | 0.115 | 0.117 | 0.122 | **0.131** |
| PTB-XL (I) | 0.069 | 0.080 | 0.070 | 0.069 | **0.082** |

## 5.3 Time Series Super-resolution

This study introduces a benchmark for super-resolution in time series data, which can be crucial for enhancing the detail and quality of data, particularly when dealing with under-sampled or low-resolution datasets. The objective of the super-resolution task is to reconstruct the original high-resolution signal from the down-sampled version and increase the temporal resolution by reconstructing the data at finer granularities. The super-resolution task differs from traditional imputation in that it focuses on upsampling the temporal resolution rather than simply filling in missing values. Specifically, it requires the model to infer the higher-frequency components that are missing due to down-sampling. For instance, reconstructing a 100 Hz ECG signal from 10, 20, 50 Hz inputs requires generating fine-grained details that are not directly observable in the low-resolution input.

In this context, we conduct super-resolution experiments on the ETT, traffic, ECG (PTB-XL), EEG (Sleep-EDFE), and electrodermal activity (EDA) datasets from the CLAS study (Markova et al., 2019). We evaluate the performance of various models, including *AdaWaveNet*, by upscaling the temporal resolution at ratios of 2, 5, 10. Our results demonstrate that AdaWaveNet excels at reconstructing high-resolution signals, capturing both low- and high-frequency components, which leads to superior performance compared to other baseline models.

Table 3 shows the evaluation results for the super-resolution task. The proposed method shows competitive results compared to the baseline methods, especially on datasets such as Traffic, Sleep-EDFE, and CLAS. We credit the observed improvement to the multi-scale analysis inside the model architecture. In this experiment, we find the frequency domain enhanced methods including FreTS, TimesNet, and FEDformer generally provide better results in strongly periodic data. For example, the first three places of averaged super-resolution reconstruction performances on ECG signals in the PTB-XL dataset are TimesNet, *AdaWaveNet*, and FreTS, respectively. The channel-wise attention also brings advantages to models, such as *AdaWaveNet* and iTransformer, in reconstructing the high-dimensional traffic data.

## 6 Discussion

In this section, we discuss the model ablations and efficiency. Also, we showcase an example of the multi-scale analysis performed by the proposed method. Further, we discuss the limitations of this work.

## 6.1 Ablation Study

An ablation study was conducted to assess the contribution of individual components within the proposed *AdaWaveNet* framework to its overall performance. This involved evaluating the model on forecasting tasks and extended imputation tasks with the omission of specific components: the Grouped Linear Module, reversible instance normalization (RevIn) as described by (Kim et al., 2021), channel-wise Attention inspired by Liu et al. (2023a), and the *AdaWaveNet* and *InvAdaWaveNet* blocks. The results are presented in Table 4, which indicates that the AWB component significantly enhances performance on the ECL and PTB-XL datasets for imputation tasks. Meanwhile, CWA shows notable efficacy in forecasting tasks. The GL module also demonstrates improved results on the PTB-XL dataset during imputation.

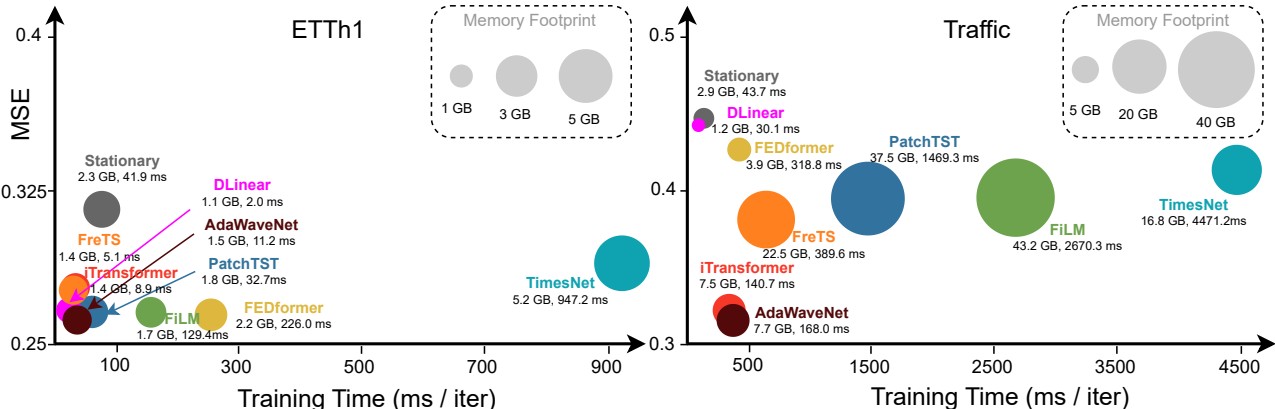

Figure 3: Comparison of model efficiency on the ETTh1 and Traffic datasets. The y-axis represents the averaged mean squared error performances across the forecasting and super-resolution tasks; whereas the x-axis represents the model training time for each iteration. The input length is 192 with a batch size of 16.

## 6.2 Efficiency Analysis

Efficiency is one of the important concerns in time series analysis. Therefore, we evaluate the efficiency of the proposed method and compare the results with all the baseline models we used in the main experiments. Controlling the input length of the sequences as 96 time points, we select the data samples from the ETTh1 and Traffic sets, as they have the least (7) and the most (862) numbers of variates. The environment used in this evaluation is AWS g5 instances with *Nvidia A10* GPUs. The batch size is fixed as 16. Following the previous studies (Wu et al., 2022; Liu et al., 2023a), we evaluate the efficiency of the models in aspects of performances (MSE), training speed, and memory footprints.

The comparison results shown in Figure 3 illustrate that *AdaWaveNet* achieves state-of-the-art performance while maintaining reasonable model efficiency. The DLinear model exhibits the smallest memory footprint and fastest training time among the evaluated models. *AdaWaveNet* demonstrates comparable training speed and memory usage to the iTransformer; while achieving significant savings in memory and training time compared to established approaches such as PatchTST.

## 6.3 Wavelet Decomposition

Figure 4 illustrates the wavelet decomposition process employed by *AdaWaveNet* in a time series forecasting task. The decomposition splits the seasonal component into layers of approximations and detail coefficients. A channel-wise attention mechanism is then applied to forecast future sequence approximations. Each decomposition level incorporates a residual connection that integrates the reconstructed signals from the *InvAdaWave* blocks. This example illustrates the multi-scale analytical capability of the proposed method, which enables the model to extract features across various granularities. Moreover, the flexibility introduced in the learnable CNN kernels empowers the *AdaWave* and *InvAdaWave* blocks to dynamically adapt to varying input data.

## 6.4 Limitations

### 6.4.1 Model Complexity

*AdaWaveNet* incorporates effective components, such as the grouped linear module and cross-channel attention mechanisms, to model dependencies across similar variates in the trend phase. It also introduces multi-scale capabilities through the *AdaWave* blocks. Despite its promising performance, *AdaWaveNet* is less efficient compared to simpler MLP-based models such as DLinear Zeng et al. (2023), which potentially limits its applicability in environments where computational resources are constrained or real-time analysis is required.

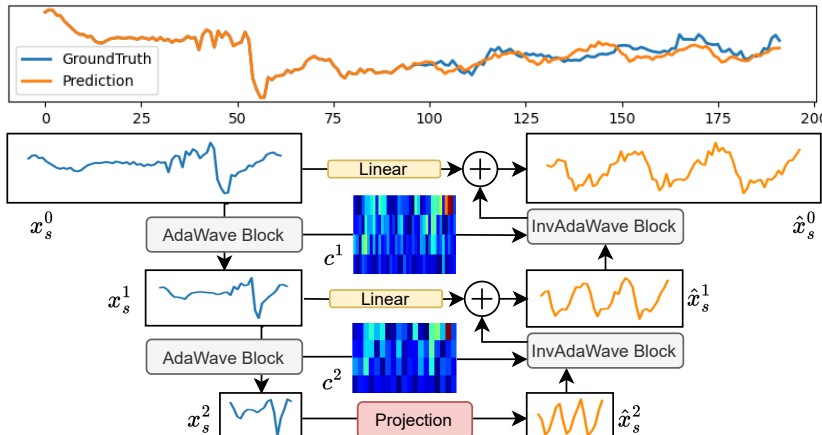

Figure 4: An example of the wavelet decomposition and projection in time series forecasting on ETTh1 data with two layers of AdaWave and InvAdaWave blocks. $x_s^l$ and $\hat{x}_s^l$ represents the the approximation at wavelet level $l$ of the input and target sequences, respectively. $c$ is the wavelet coefficients decomposed by the *AdaWave* blocks.

### 6.4.2 Generalization to Different Signal Types

*AdaWaveNet* demonstrates robust performance in forecasting tasks with data types such as weather and solar, as well as in extended imputation tasks with electricity and EEG data. However, its capacity to generalize across the full spectrum of time series data and various tasks has room to be further improved. For example, in tasks involving random imputation of electricity and weather datasets, TimesNet exhibits superior performance. Similarly, iTransformer outperforms *AdaWaveNet* in traffic forecasting tasks.

## 7 Conclusion and Future Work

This paper presented *AdaWaveNet*, a novel and efficient architecture for addressing the challenges of non-stationarity in time series data analysis. Through the integration of adaptive wavelet transformations, *AdaWaveNet* demonstrates the advantages of modeling multi-scale data representations in time series data. Our extensive experiments are conducted on 10 datasets for forecasting, imputation, and the newly established super-resolution tasks, and the results indicate the effectiveness of our approach.

Future efforts will focus on exploring and expanding *AdaWaveNet* and its multi-scale analysis framework to a broader range of tasks, such as classification and anomaly detection. Also, we aim to explore the possibility of using multi-scale analysis in large-scale pre-training models.

### Acknowledgments

This work was supported by National Science Foundation (# 2047296) and National Institute of Health (# R01DA059925)

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

# Appendix

# A    Implementation Details

In this section, we cover the implementation details of the methods and experiments, including datasets, algorithm details, hyperparameters, and baselines.

## A.1    Datasets

There are 10 datasets used in this study, and the overall summary of the datasets information can be seen in Table 5. Also, the descriptions of each set are introduced in this section.

**ETT (Electricity Transformer Temperature):** The ETT dataset (Zhou et al., 2021) includes detailed records of electricity transformer temperatures and corresponding electricity consumption data. This high-resolution dataset, sampled every 15 minutes, covers two years. It provides insights into the relationship between transformer temperatures and electricity demand, crucial for maintaining efficient and safe operations of power systems. In our research, we utilize the ETT dataset to forecast short-term electricity demand and transformer temperatures.

**Electricity Load Diagrams:** The Electricity Load Diagrams dataset, sourced from the UCI Machine Learning Repository (Asuncion & Newman, 2007), comprises electricity consumption data recorded at 15-minute

Table 5: Details of datasets used in the experiments. Data Split means the number of samples split into the train, validation, and test sets.

| Dataset | Dimentions | Frequency | Data Split | Forecasting | Imputation | SR |
|---|---|---|---|---|---|---|
| ETTm1 | 7 | 15 mins | (34465, 11521, 11521) | ✓ | | ✓ |
| ETTh1 | 7 | Hourly | (8545, 2881, 2881) | ✓ | | ✓ |
| ECL | 321 | Hourly | (18317, 2633, 5261) | ✓ | ✓ | |
| Traffic | 862 | Hourly | (12185, 1757, 3509) | ✓ | | ✓ |
| Weather | 21 | 10 mins | (36792, 5291, 10540) | ✓ | ✓ | |
| Exchange | 8 | Daily | (5120, 665, 1422) | ✓ | | |
| Solar | 137 | 10 mins | (36601, 5161, 10417) | ✓ | | |
| PTB-XL | 12 | 500 Hz | (14771, 1493, 1652) | | ✓ | ✓ |
| Sleep-EDFE | 1 | 100 Hz | (22212, 9519, 10577) | | ✓ | ✓ |
| CLAS | 1 | 100 Hz | (993, 0, 359) | | | ✓ |

intervals from multiple customers. The dataset represents a diverse range of electricity consumers, including both individual households and industrial customers. The data spans from 2011 to 2014, encompassing various consumption patterns and seasonal effects. For our study, we focus on forecasting electricity demand on an hourly basis.

**Traffic (PeMS: Performance Measurement System):** The PeMS dataset, provided by the California Department of Transportation, is a comprehensive collection of traffic flow data. It contains real-time traffic speed and volume information collected from over 39,000 individual sensors across the freeway system of California. These sensors report data every 30 seconds, offering an exceptionally detailed view of traffic patterns. Our study utilizes this dataset to forecast traffic flow and congestion levels. The data spans several years, but for our analysis, we focus on a one-year period, ensuring a mix of workdays and weekends to capture varying traffic behaviors.

**Weather (Global Surface Summary of the Day):** The weather dataset from NOAA's National Climatic Data Center (Wetterstation, 2020) offers daily weather summaries from a wide array of weather stations around the world. This dataset includes essential meteorological parameters such as temperature, humidity, precipitation, wind speed, and atmospheric pressure. The data, spanning over several decades, provides a rich source for analyzing and forecasting weather patterns. For our research, we select a subset of the dataset encompassing ten years of data from stations across different climatic zones. The goal is to develop models capable of predicting weather conditions such as temperature and precipitation.

**Exchange Rates:** The Foreign Exchange Rates dataset (Lai et al., 2018) encompasses daily exchange rates of various currencies against the US dollar from the year 2000 to 2019. It is a comprehensive dataset that includes the exchange rates of over 50 currencies and offers a detailed view of global financial dynamics. Our study aims to forecast the daily exchange rates of major currencies. The dataset's span allows for the analysis of long-term trends as well as responses to major global events. For preprocessing, we ensure data continuity by addressing any missing values and then normalize the data to account for different scales across currencies.

**Solar Energy Prediction:** The Solar Energy Prediction dataset from the UCI Machine Learning Repository (Asuncion & Newman, 2007) contains solar power generation data alongside various meteorological variables. The dataset, collected over one year, includes measurements such as solar irradiance, temperature, humidity, and wind speed, sampled at 10-minute intervals. This comprehensive dataset enables the development of models for predicting solar energy output, a key factor in managing renewable energy resources. In our analysis, we focus on forecasting daily solar energy data.

**PTB-XL:** The PTB-XL (Wagner et al., 2020) dataset is a large dataset containing 21,837 clinical 12-lead electrocardiogram (ECG) records from 18,885 patients of 10-second length, where 52% are male and 48% are female with ages ranging from 0 to 95 years (median 62 and interquartile range of 22). There are two sampling rates: 100 Hz and 500 Hz, available in the dataset, but in our experiments, only data sampled at 100 Hz are used. The raw ECG data are annotated by two cardiologists into five major categories, including normal ECG (NORM), myocardial infarction (MI), ST/T Change (STTC), Conduction Disturbance (CD), and Hypertrophy (HYP). The dataset contains a comprehensive collection of various co-occurring pathologies and a large proportion of healthy control samples. We experimented with imputation and super-resolution tasks. Further, to ensure a fair comparison of machine learning algorithms trained on the dataset, we follow the recommended splits of training and test sets, which results in a training/testing ratio of 8/1.

**Sleep-EDFE:** The Sleep-EDF (expanded) (Kemp et al., 2000) dataset contains whole-night sleep recordings from 822 subjects with physiological signals and sleep stages that were annotated manually by well-trained technicians. In this dataset, the physiological signals, including Fpz-Cz/Pz-Oz electroencephalogram (EEG), electrooculogram (EOG), and chin electromyogram (EMG), were sampled at 100 Hz. To model the relationship between the sleep patterns and physiological data, we split the whole-night recordings into 30-second Fpz-Cz EEG segments as in Supratak & Guo (2020), which resulted in a total of 42308 EEG and sleep pattern pairs. We divided 25% of the samples into a testing set according to the order of the subject IDs.

**CLAS:** The CLAS dataset Markova et al. (2019) aims to support research on the automated assessment of certain states of mind and emotional conditions using physiological data. The dataset consists of synchronized recordings of ECG, photopletysmogram (PPG), electrodermal activity (EDA), and acceleration (ACC) signals.

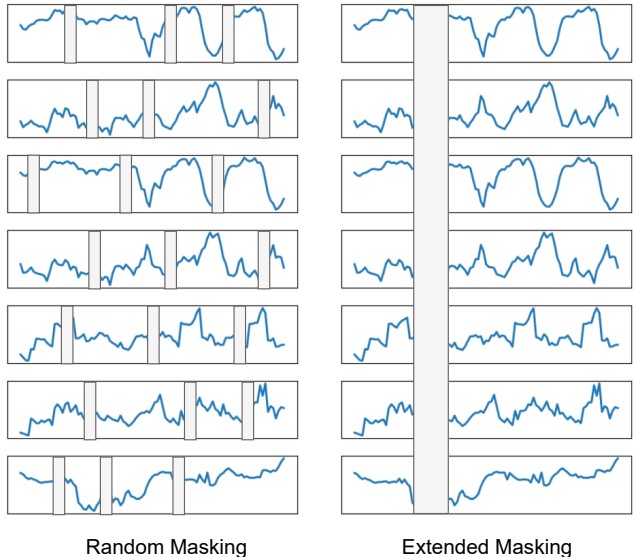

Figure 5: Comparison of masking methods for time series imputation. The left column depicts random masking, where individual data points are randomly concealed throughout the series, simulating sporadic data loss. The right column illustrates extended masking, where contiguous segments are masked to emulate prolonged periods of missing data. Each row represents an independent time series sample subject to the respective masking method.

Sixty-two healthy subjects participated and were involved in three interactive tasks and two perceptive tasks. The perceptive tasks, which leveraged the images and audio-video stimuli, were purposely selected to evoke emotions in the four quadrants of arousal-valence space. In this study, our goal was to use the EDA signal to detect binary high/low stress states that are annotated in arousal-valence space. We processed the raw EDA data with a lowpass Butterworth filter with a cutoff frequency of 0.2 Hz, then split the sequences into 10-second segments. We divided the train/test set in a subject-independent manner and utilized the data from 17 subjects as the test set according to subject ids ($> 45$).

## A.2 Imputation Settings

In the context of time series imputation, masking refers to the process of artificially removing or concealing parts of the data to simulate missing values, which the model then attempts to impute. Two distinct masking approaches are commonly employed: random masking and extended masking.

**Random Masking**: This method involves randomly selecting points or segments within the time series data and setting them as missing or masked. The randomness of this approach is intended to simulate data missingness that occurs sporadically and without a specific pattern, which is a common scenario in real-world datasets. However, this method may not adequately represent scenarios where data is missing for extended periods, which is also a practical occurrence.

**Extended Masking**: To address the limitations of random masking, we introduce the extended masking approach. This technique creates artificial gaps by masking contiguous segments of the time series. Extended masking is designed to simulate scenarios where data might be missing due to prolonged outages or systematic issues, providing a more challenging and realistic task for the imputation model to tackle.

Both methods are essential for testing the robustness and versatility of imputation models, ensuring they can handle various types of missing data patterns.

Table 6: Comparison of performances of using different numbers of clusters in grouped linear module.

| Dataset / number of clusters | $k = 1$ | $k = 2$ | $k = 3$ | $k = 4$ | $k = 5$ |
|---|---|---|---|---|---|
| ECL | 0.175 | 0.173 | 0.170 | **0.168** | 0.169 |
| Weather | 0.240 | **0.233** | 0.236 | 0.235 | 0.235 |
| Exchange | **0.481** | 0.484 | 0.482 | 0.481 | 0.482 |
| Solar | **0.209** | 0.214 | 0.210 | 0.212 | 0.214 |
| ETTh1 | 0.462 | 0.457 | 0.449 | **0.444** | 0.446 |

Table 7: The hyperparameter settings of *AdaWaveNet* for each dataset.

| hyperparam. | lifting kernel size | lifting Levels | n_clusters | learning rate |
|---|---|---|---|---|
| ETT | 7 | 4 | 4 | 0.0005 |
| ECL | 7 | 3 | 4 | 0.0005 |
| Traffic | 5 | 1 | 9 | 0.001 |
| Exchange Rate | 7 | 4 | 1 | 0.0001 |
| Weather | 7 | 4 | 2 | 0.0005 |
| Solar | 7 | 1 | 1 | 0.0005 |
| PTB-XL | 16 | 5 | 2 | 0.001 |
| Sleep-EDFE | 16 | 5 | 1 | 0.001 |
| CLAS | 9 | 5 | 1 | 0.001 |

### A.3 Implementation Details

In this section, we introduce the experimental environments and the hyperparameter we used for each dataset

### A.3.1 Experimental Environments

All the modules are implemented in PyTorch 1.11 and a Python version of 3.10. To speed up the experiments, GPU instances are utilized. We use AWS G5 instances, which are equipped with Nvidia A10 24GB GPUs, in training and inferring the models.

### A.3.2 Hyperparameter Setting

In this section, we list some essential hyperparameters used in this study. For the design of *AdaWave* blocks, we adjust the depth of transformations in a range of 1 to 5; whereas the kernel size of the utilized convolutional kernels is adjusted based on the datasets. The number of clusters used in the grouped linear model varies from 1 to 9 depending on the channels of signals. Also, we adjust the learning rate for each set of experiments to achieve better convergent performances. Further, to determine the number of clusters in the proposed group linear module, we conduct extensive experiments as shown in Table 6. The detailed hyperparameters of each dataset are listed in Table 7.

### A.4 AdaWaveNet Ablation Study

Table 8 shows the full results of the ablations of the model.

## B Case Study: Analyses of Non-stationary Signals with *AdaWaveNet*

In this case study, we present a comprehensive approach to synthesizing non-stationary time series data, which combines low-frequency sinusoidal components, high-frequency transients, linear trends, and Gaussian noise to create a basic non-stationary signal. This is represented by the equation:

$$s(t) = \sin(2\pi f_1 t) + \sin(2\pi f_2 t)e^{-\alpha(t-t_0)^2} + \beta t + \varepsilon(t) \tag{14}$$

Table 8: The full results of model ablation with mean squared error as the evaluation metric. The highest MSE is highlighted in bold, while the second highest is underlined. Refer to Table 8 in the Appendix for comprehensive results.

| | | w/o | / | | Grouped Linear | | RevIn | | Channel Attention | | AdaWave Block | |
|---|---|---|---|---|---|---|---|---|---|---|---|---|
| | Metrics | MSE | MAE | MSE | MAE | MSE | MAE | MSE | MAE | MSE | MAE |
| Forecasting | Weather 96 | 0.169 | 0.215 | 0.174 | 0.226 | 0.180 | 0.231 | 0.185 | 0.239 | 0.177 | 0.231 |
| | 192 | 0.203 | 0.245 | 0.209 | 0.254 | 0.216 | 0.260 | 0.227 | 0.266 | 0.222 | 0.257 |
| | 336 | 0.248 | 0.286 | 0.255 | 0.299 | 0.262 | 0.311 | 0.272 | 0.328 | 0.276 | 0.337 |
| | 720 | 0.313 | 0.336 | 0.321 | 0.348 | 0.325 | 0.356 | 0.341 | 0.375 | 0.345 | 0.382 |
| | Avg. | 0.233 | 0.271 | 0.240 | 0.282 | 0.246 | 0.290 | **0.256** | **0.302** | 0.255 | 0.302 |
| | Traffic 96 | 0.417 | 0.291 | 0.429 | 0.303 | 0.418 | 0.293 | 0.513 | 0.338 | 0.472 | 0.314 |
| | 192 | 0.401 | 0.281 | 0.415 | 0.297 | 0.409 | 0.288 | 0.499 | 0.331 | 0.466 | 0.312 |
| | 336 | 0.407 | 0.284 | 0.416 | 0.297 | 0.417 | 0.292 | 0.520 | 0.349 | 0.470 | 0.306 |
| | 720 | 0.433 | 0.297 | 0.439 | 0.314 | 0.442 | 0.314 | 0.545 | 0.382 | 0.483 | 0.317 |
| | Avg. | 0.415 | 0.288 | 0.425 | 0.303 | 0.422 | 0.297 | **0.519** | **0.350** | 0.473 | 0.312 |
| Imputation | ECL 0.125 | 0.098 | 0.207 | 0.099 | 0.208 | 0.103 | 0.212 | 0.108 | 0.214 | 0.112 | 0.219 |
| | 0.25 | 0.104 | 0.207 | 0.106 | 0.212 | 0.106 | 0.214 | 0.112 | 0.215 | 0.121 | 0.227 |
| | 0.375 | 0.121 | 0.228 | 0.124 | 0.233 | 0.129 | 0.240 | 0.131 | 0.232 | 0.140 | 0.240 |
| | 0.5 | 0.126 | 0.230 | 0.131 | 0.237 | 0.130 | 0.235 | 0.137 | 0.243 | 0.149 | 0.250 |
| | Avg. | 0.112 | 0.218 | 0.115 | 0.223 | 0.117 | 0.225 | 0.122 | 0.226 | **0.131** | **0.234** |
| | PTB-XL 0.125 | 0.049 | 0.062 | 0.058 | 0.071 | 0.052 | 0.065 | 0.053 | 0.064 | 0.063 | 0.084 |
| | 0.25 | 0.063 | 0.080 | 0.072 | 0.089 | 0.067 | 0.088 | 0.062 | 0.078 | 0.072 | 0.091 |
| | 0.375 | 0.074 | 0.089 | 0.085 | 0.114 | 0.071 | 0.082 | 0.077 | 0.089 | 0.090 | 0.110 |
| | 0.5 | 0.089 | 0.122 | 0.103 | 0.136 | 0.091 | 0.121 | 0.085 | 0.117 | 0.104 | 0.127 |
| | Avg. | 0.069 | 0.088 | 0.080 | 0.103 | 0.070 | 0.089 | 0.069 | 0.087 | **0.082** | **0.103** |

where $f_1$ and $f_2$ are the low and high frequencies respectively, $\alpha$ controls the transient decay, $\beta$ represents the linear trend, and $\varepsilon(t)$ is Gaussian noise. This formulation captures fundamental non-stationary behaviors while maintaining interpretability. We first generate signals with a regular low-frequency component (5 Hz) and a high-frequency component (50 Hz) as the simple signal. Further, for increased complexity, we introduce traffic-like signals that incorporate daily and weekly cyclical patterns to mimic real-world traffic fluctuations:

$$s_{\text{traffic}}(t) = \sin(2\pi 24t) + 0.5\sin(4\pi 24t) + \sin(2\pi 7t) + 0.5t + \varepsilon(t) \tag{15}$$

Similarly, electricity consumption-mimic signals simulate seasonal and daily variations typical of power usage data:

$$s_{\text{electricity}}(t) = 5\sin(2\pi t) + 2\sin(4\pi t) + 3\sin(2\pi 365t) + 2t + \varepsilon(t) \tag{16}$$

Both complex models include trend components and noise to simulate realistic non stationary data. Furthermore, we implement optional variance shifts and step changes to test model robustness against abrupt signal changes. The variance shift is modeled as an increase in the amplitude of $\varepsilon(t)$ at a specific time point, while the step change is represented by a sudden additive constant to the signal. Based on the three settings - including simple, traffic, and electricity - we further adjust (1) the variance shift values to be 1 and 2, for moderate and large shifts, (2) the step changes as -0.5 or 0.5 to indicate the negative and positive changes. The synthetic signals are visualized in Figure 6. In total, for each signal, 1024 data points are generated, where the first 512 timestamps are used in training models. In the evaluation stage, we compare model predictions with a denoised signal, i.e., $\sin(2\pi \cdot 5t) + \sin(2\pi \cdot 50t) \cdot e^{-50(t-0.5)^2} + t$. Also, the variance shifts and step changes are only applied in the test phases.

The performance of various models, including *AdaWaveNet*, DLinear, TimesNet, FreTS, PatchTST, and iTransformer, was evaluated on this synthetic non-stationary time series. Table 9 summarizes the results of this case study. Further, we showcase two examples, including Simple and Traffic with variance shift and step changes, in Figure 7 and 8. With the lowest MSE and MAE among all models, *AdaWaveNet*'s predictions closely align with the ground truth, effectively capturing both the low-frequency sinusoidal component and the high-frequency transient component. In contrast, other models struggle to varying degrees in accurately predicting the non-stationary signal. DLinear shows significant deviation from the ground truth, particularly in capturing the high-frequency components, which results in high MSE and MAE. FreTS and iTransformer demonstrate moderate performance but fail to capture the finer details of the signal, particularly the high-frequency components.

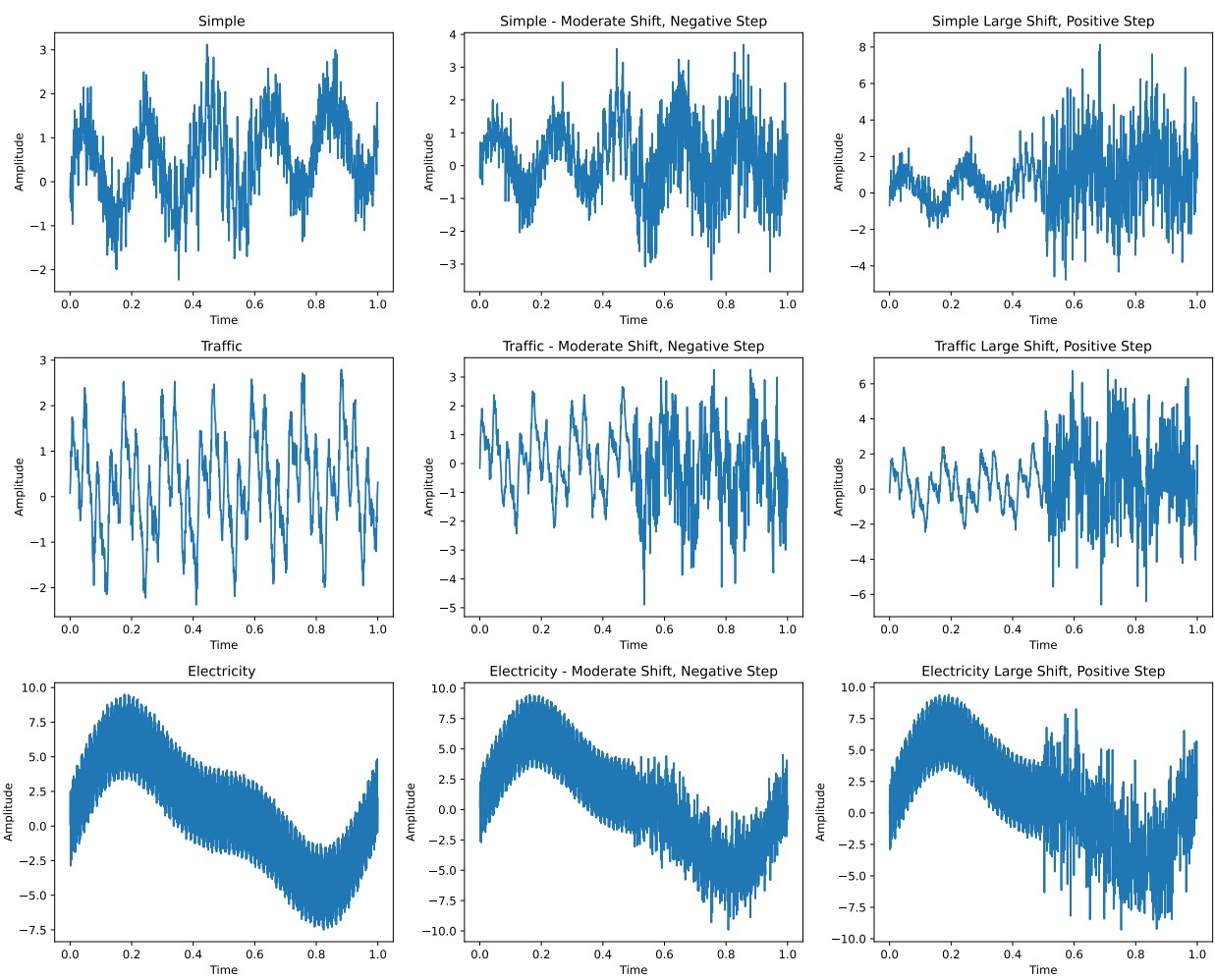

Figure 6: Synthetic non-stationary time series data generated using three complexity levels (Simple, Traffic, and Electricity) and three parameter settings (No shift/step, Moderate shift with negative step, and Large shift with positive step). Each row represents a different complexity level, while each column shows the conditions of variance shift and step changes.

Table 9: The averaged results of the model forecasting on synthetic non-stationary signals. Models are trained on 512 data samples, whereas the test samples are the later 512 points. Both the lookback length and prediction length are fixed at 96. The evaluation metrics are mean squared error (MSE) and mean absolute error (MAE)

| Models | Simple | | Traffic | | Electricity | |
|---|---|---|---|---|---|---|
| | MSE | MAE | MSE | MAE | MSE | MAE |
| iTransformer | 0.454 | 0.525 | 1.268 | 0.904 | 2.677 | 2.564 |
| DLinear | 0.599 | 0.692 | 1.032 | 0.858 | 1.885 | 1.679 |
| TimesNet | 0.397 | 0.492 | 1.622 | 1.027 | 2.571 | 2.488 |
| FreTS | 0.676 | 0.707 | 2.081 | 1.169 | 1.701 | 1.502 |
| PatchTST | 0.330 | 0.453 | 1.168 | 0.866 | 1.550 | 1.351 |
| AdaWaveNet | **0.215** | **0.367** | **0.478** | **0.576** | **1.299** | **1.193** |

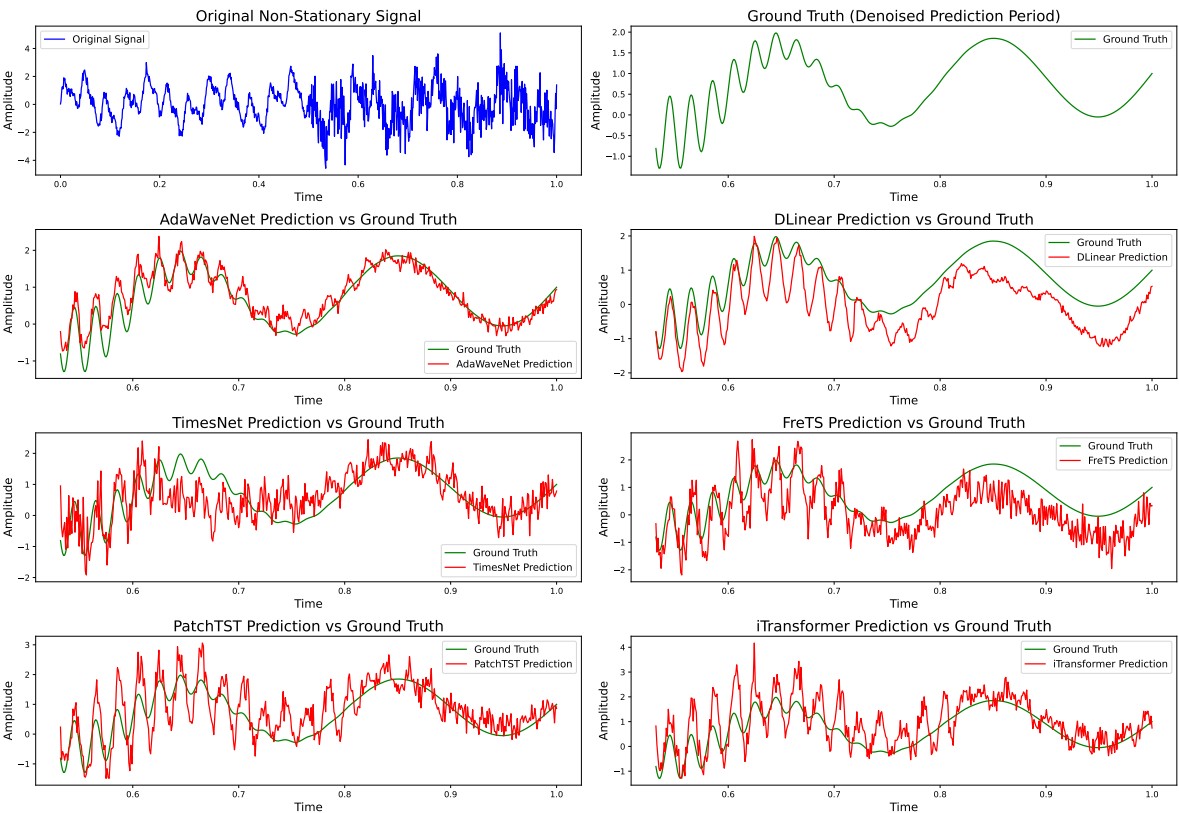

Figure 7: Comparison of model predictions on a synthetic non-stationary time series [Simple]. Top row: Original non-stationary signal (left) and ground truth for the prediction period (right). Subsequent rows: Predictions from AdaWaveNet, DLinear, TimesNet, FreTS, PatchTST, and iTransformer models compared against the ground truth.

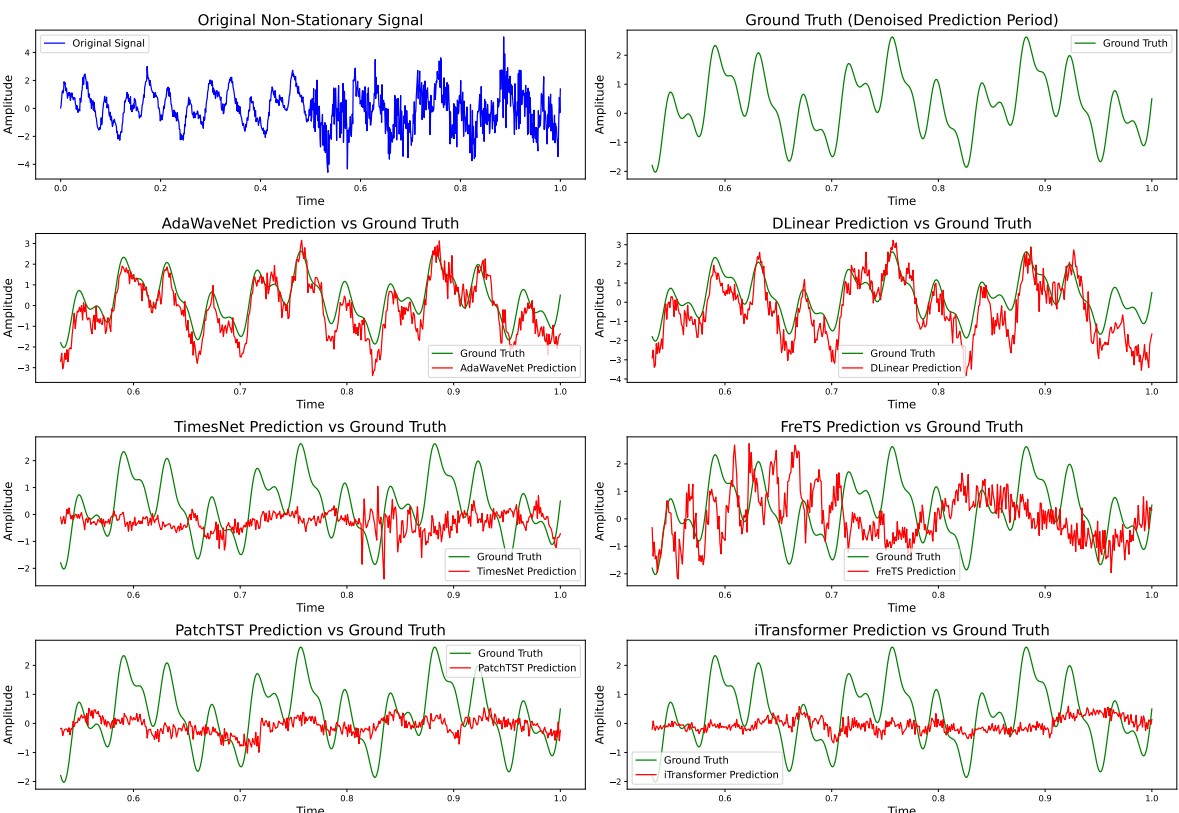

Figure 8: Comparison of model predictions on a synthetic non-stationary time series [Traffic - moderate variance shift & negative step change]. Top row: Original non-stationary signal (left) and ground truth for the prediction period (right). Subsequent rows: Predictions from AdaWaveNet, DLinear, TimesNet, FreTS, PatchTST, and iTransformer models compared against the ground truth.

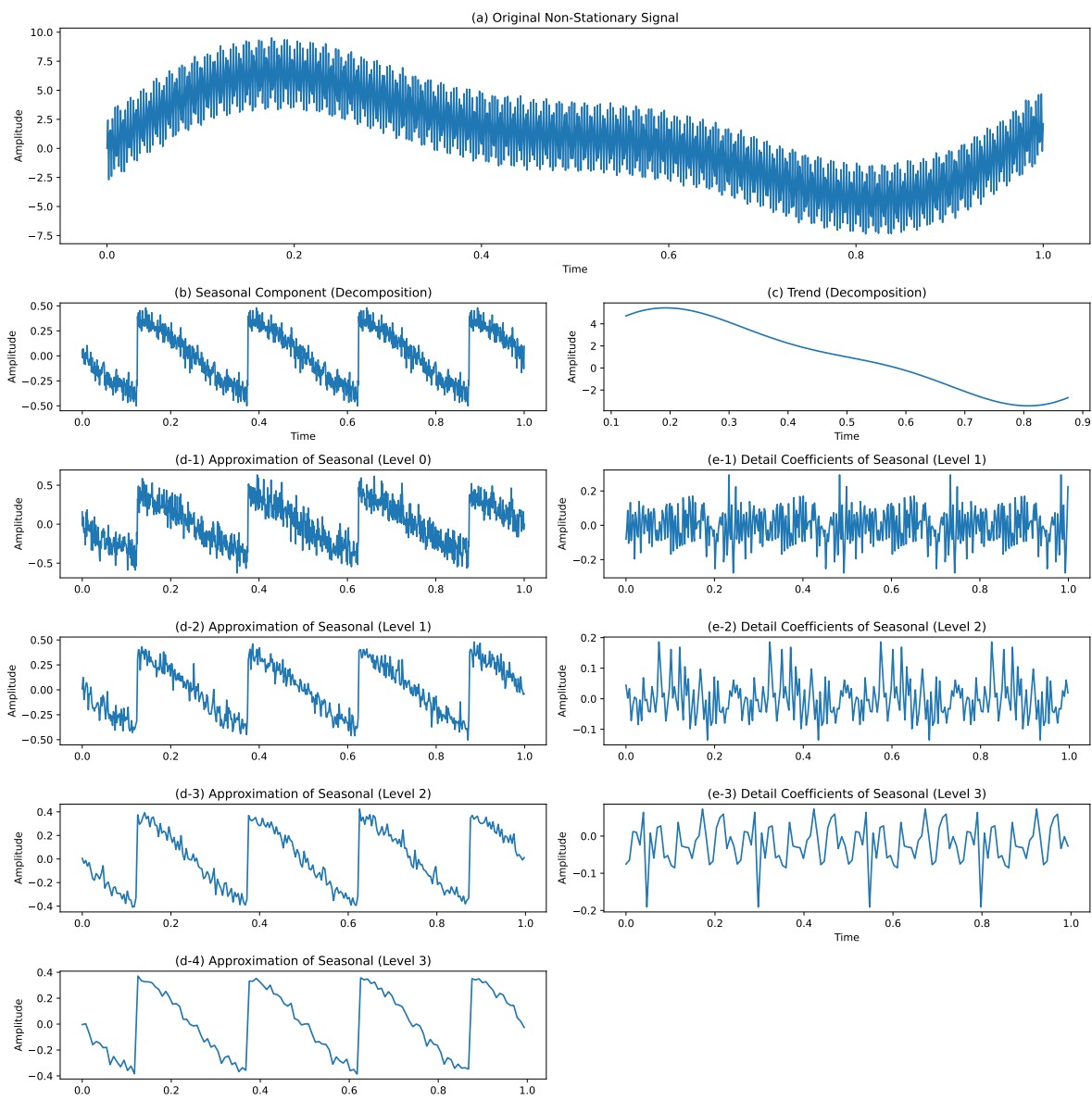

Figure 9: Multi-level decomposition of a non-stationary time series using AdaWaveNet. The top plot shows the input signal, followed by four levels of approximations (left column) and their corresponding wavelet coefficients (right column). Each level represents a coarser approximation of the original signal, with the coefficients capturing the details lost between successive approximation levels.

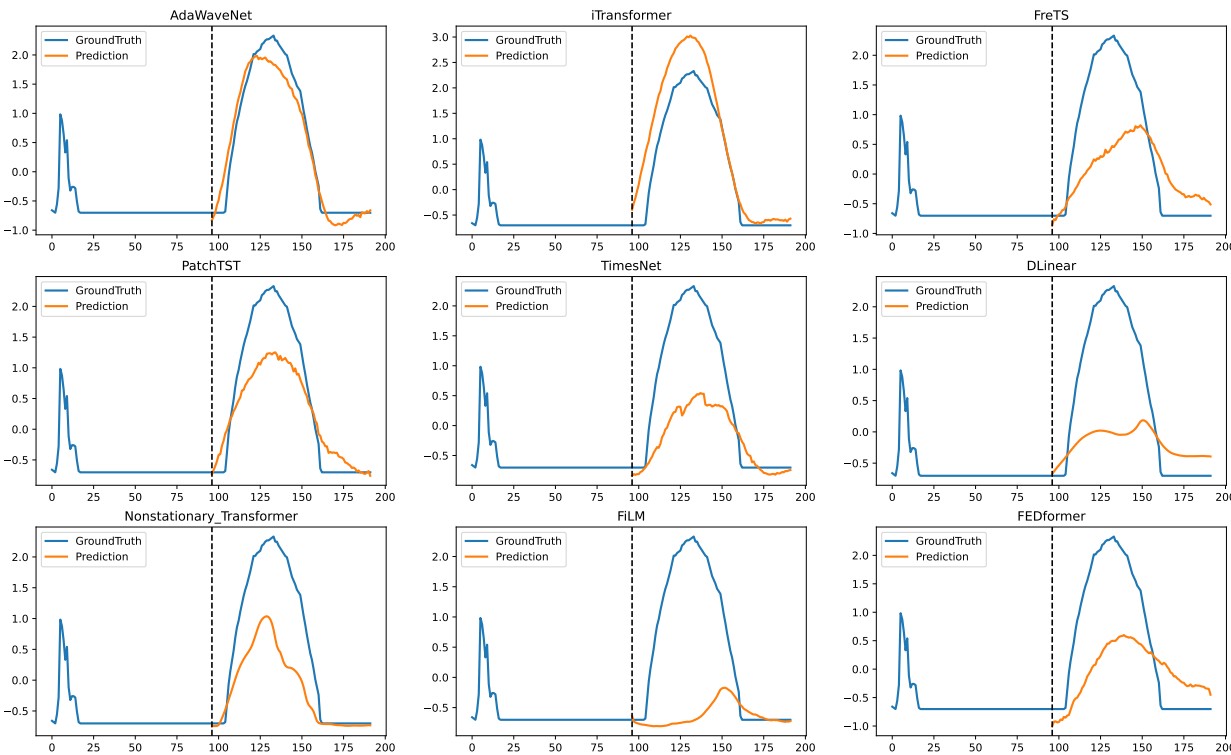

Figure 10: Visualization of a forecasting task on traffic dataset. The length of both the input sequence and forecasting sequence is 96.

Figure 9 illustrates the multi-scale decomposition capabilities of *AdaWaveNet* when processing the generated non-stationary time series. The approximation at each level represents a progressively coarser version of the original signal, which effectively separates low-frequency trends from high-frequency details. Concurrently, the wavelet coefficients encode the fine-grained information lost between successive approximation levels. This hierarchical decomposition allows *AdaWaveNet* to adapt to the signal's changing characteristics at multiple scales. For instance, the Level 0 approximation retains much of the original signal's structure, while Levels 1, 2, and 3 focus on broader trends. This adaptive multi-scale approach enables the proposed model to learn both local fluctuations and overarching patterns in non-stationary time series.

## C  Showcases

In this section, we showcase some examples of forecasting, imputation, and super-resolution tasks.

### C.1  Forecasting

Figure 10 presents a comparative example of forecasting future traffic volumes using various models. The figure reveals a notable disparity between past sequences and predicted sequences for this particular variate. The visual results indicate that *AdaWaveNet* yields the most accurate forecast in this instance. Additionally, the iTransformer model also performs commendably, which suggests that its channel-wise attention mechanism is particularly useful for analyzing past traffic patterns for prediction purposes.

### C.2  Imputation

The examples of imputation, including the random imputation and extended imputation are demonstrated in this section.

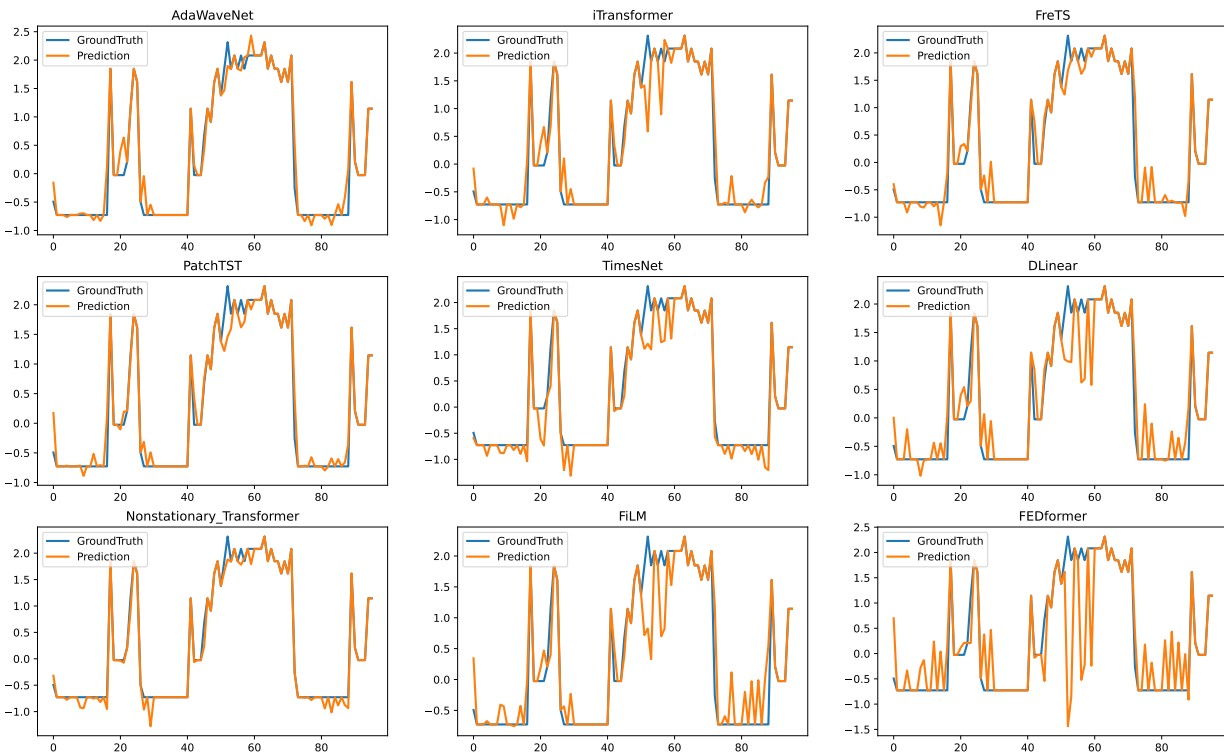

Figure 11: Visualization of a random imputation task on ETTh1 dataset. The sequence length is 96 and the masked ratio is 0.25.

### C.2.1 Random Masking

We provide an example of imputation with the random masking method. The proposed *AdaWaveNet* and all the other baseline methods. As shown in Figure 11, AdaWaveNet, alongside baseline methods such as PatchTST, TimesNet, and Nonstationary-Transformer, is capable of capturing the temporal dynamics. Notably, AdaWaveNet shows superior performance in imputing fine-grained details, effectively handling both the seasonality during peak phases and the underlying trends in flatter regions.

### C.2.2 Extended Masking

We provide an example of imputation with the extended masking method. The proposed *AdaWaveNet* and all the other baseline methods. Shown as in Figure 12, the proposed method exhibits a close approximation to the ground truth, which indicates a higher predictive accuracy within this interval. The consistency across the models outside the masked region implies a shared ability to capture the temporal dynamics in non-masked intervals; while the differences within the masked region highlight the distinct predictive capabilities and potential overfitting issues of the individual models.

### C.3 Super-resolution

The visualization presents the results of a super-resolution task on time series data, specifically forecasting traffic volume. Each subplot represents the performance of a different model: *AdaWaveNet*, iTransformer, FreTS, TimesNet, DLinear, PatchTST, Non-stationary Transformer, FiLM, and FEDformer. In each plot, three lines are denoted as the Ground Truth (blue line), which is the actual high-resolution data; the Prediction (orange line), which is the model's predicted high-resolution data; and the Low-resolution Input (gray line), which serves as the model's input data and represents the downsampled or coarse version of the Ground Truth.

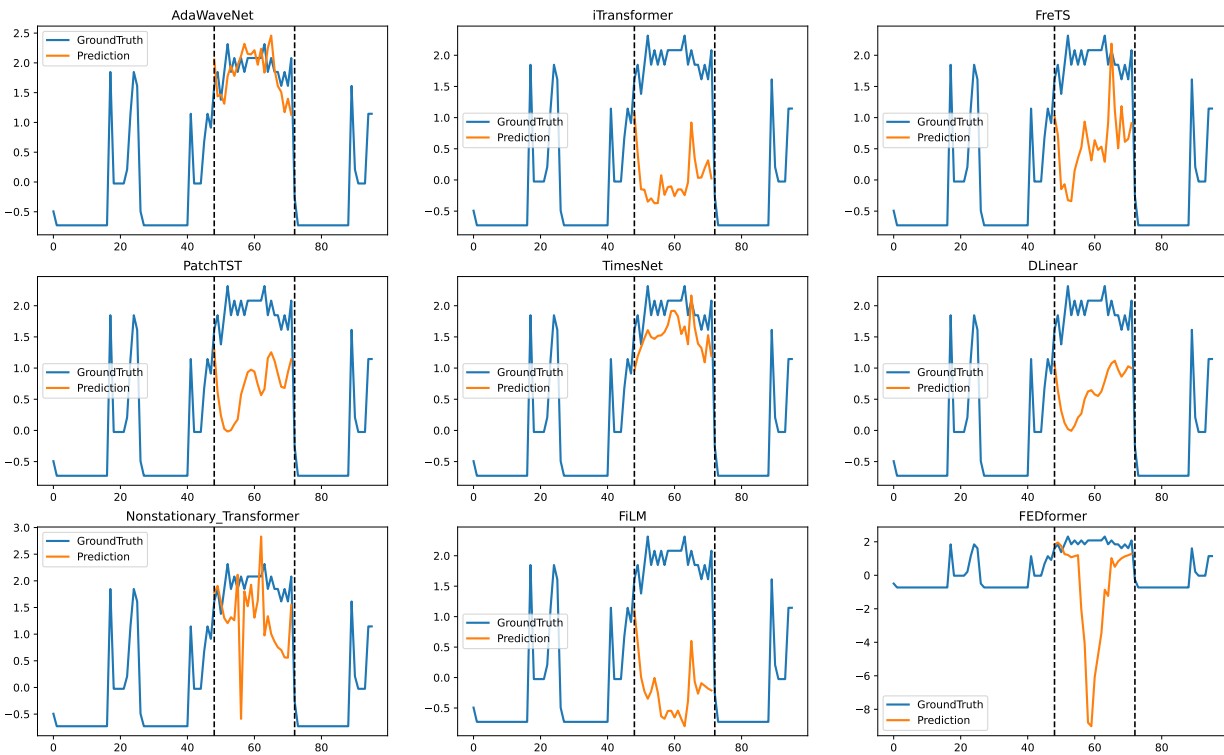

Figure 12: Visualization of an extended imputation task on ETTh1 dataset. The sequence length is 96 and the masked ratio is 0.25.

The predictions of *AdaWaveNet* closely follow the Ground Truth across the entire sequence. The fidelity of *AdaWaveNet*'s prediction to the Ground Truth, particularly in capturing the peaks and troughs of the traffic volume, showcases the model's capability in the super-resolution task. The granularity of details in the prediction suggests that *AdaWaveNet* effectively upsamples the low-resolution input and reconstructs nuanced and accurate traffic data.

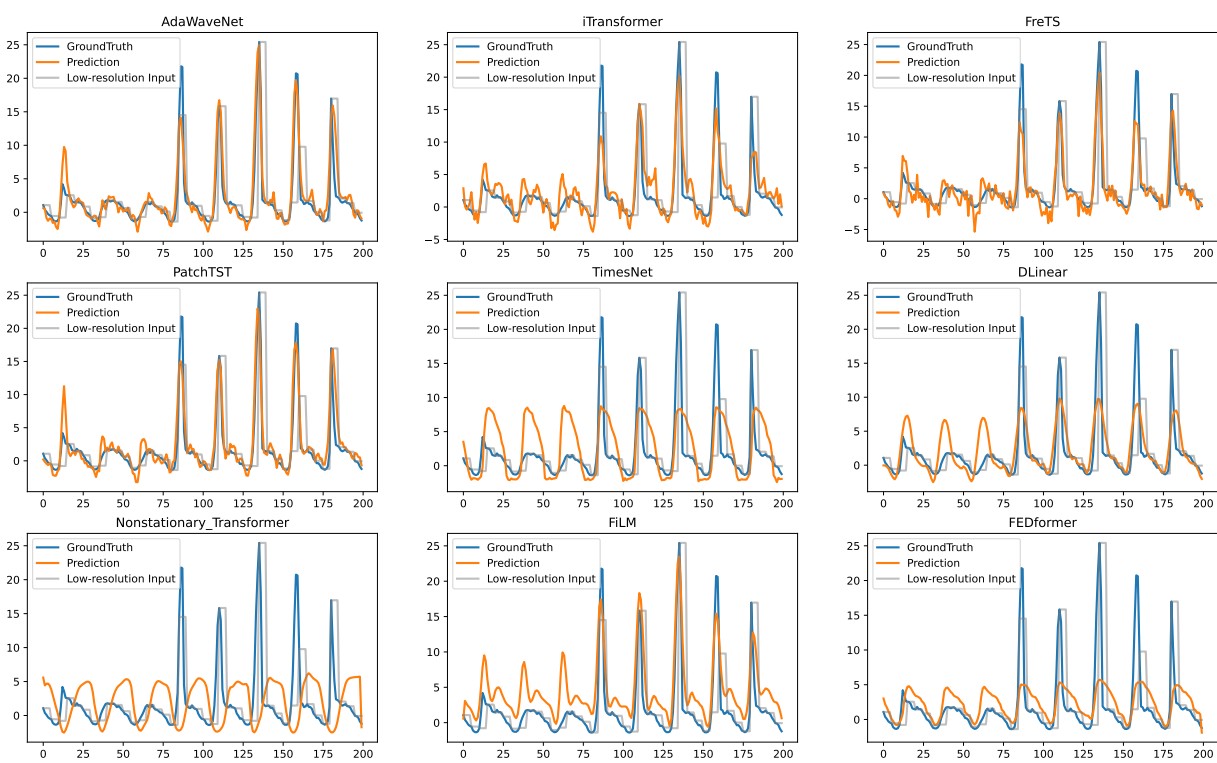

Figure 13: Visualization of a super-resolution task on traffic data with an input length of 200. The super-resolution ratio is 5.

