# OpenReview forum: "AdaWaveNet: Adaptive Wavelet Network for Time Series Analysis"
_TMLR — Accepted by TMLR_

### Review · Reviewer_Du9K · 2024-07-02

**Summary Of Contributions:**

The manuscript considers time serial analysis, specifically cases in which signal are not stationary. They propose a new architecture to model such time series based on Wavelets and Lifting scheme. They apply their proposed approach to several non-stationary time series estimation tasks including forecasting, imputation and super-resolution tasks.

**Audience:**

Yes

**Claims And Evidence:**

Yes

**Requested Changes:**

Presentation should be improved as mentioned above. They could consider the following comments/questions:
- Wavelet is described in (1) as a continuous signal (f is the function/signal and x is the variable), but this seems to change in section 3.1 where x is a discrete signal. Also cij in (1) are Wavelet coefficients i.e. scalars, but in section 3.2 c (“approximation”) seems to be signal. Are c’s in (1) and section 3.2 still somehow related, so is some kind of form of Wavelet coefficients?
- Organize the experiments section as mentioned above
- I find it very odd that limitations of the method are placed in appendix.

Minor changes:
- Citations has mistake, I think in some places \citep should be used instead \cite and vice versa.
- Font in tables is very small, very hard to read if the manuscript is printed (may be general issue in latex template, the font overall is very small)
- Table captions say that lowest MSE is highlighted with red, but also MAE is highlighted. Why blue is underlined, but red is not?

**Strengths And Weaknesses:**

Strenghs:
- The approach seems interesting.
- Results looks impressive.

Weaknesses:
- Description of method and its motivation could be improved. For example, from sections 3 and 4, I can understand their idea in somewhat in high level, but it is hard to get deeper understanding about the approach. Maybe it is my lack of understanding related to lifting scheme which is new to me, but somehow I find the description is confusing and it is not clear to me that why the results architecture is “Wavelet” or motivated by it. See below.
- Results section is not very well organized. The content is some what randomly placed between main body and appendix. I would say that a general rule about appendices is that you should be able to read the manuscript without it and still get essential parts of what has been done in the research. This does not clearly hold with this manuscript as understanding of experiments requires lots of jumping back and worth between the main body and appendix. And all necessary details are not even referred in the main body, so you cannot find those without reading through the whole thing.

---

> ### Author Response · Authors · 2024-10-11
> **Thank you for the insightful comment**
>
> Thank you for your thorough review and valuable feedback on our paper. We have carefully addressed your comments:
>
> 1. **Clarifying Method Description and Motivation**:
>
>     We appreciate your feedback regarding the description of the method and its motivation. In response, we have improved the clarity of Sections 3 and 4, especially in explaining the lifting scheme and the role of wavelet transformations in our architecture.
>
>     - Section 3.1 (Wavelet Transform): We have clarified the distinction between continuous and discrete signals. Initially, we introduce wavelets in the continuous domain to describe the general concept, and then we transition to the discrete version of the wavelet transform in Section 3.1, where signals are sampled. This is done to align with practical implementations in time series data. We have now emphasized this transition more clearly.
>
>     - Relation Between $c_{i,j}$ and Approximation Coefficients (Section 3.2): We have revised the text to clarify that the $c_{i,j}$ in Equation (1) refers to the wavelet coefficients that capture signal features at different scales, whereas the approximation coefficients discussed in Section 3.2 represent the low-frequency components of the signal after wavelet decomposition. In the revised version, we improve the clarification by changing $c_{i,j}$ into $w$ for indicating the wavelet coefficients.
>
>     - Motivation for Using Lifting Scheme in Model Architecture: We have added further clarification and motivation in Section 3.2 to explain the lifting scheme and why it is important in building our model. Essentially, the lifting scheme is used to adaptively decompose time -series into multi-scale components, which brings advantages in handling read-world time series signals.
>
> 2. **Organizing the Results Section**:
>
>     We understand the reviewer’s concern regarding the organization of the results and the reliance on the appendix. In response, we have moved some of the key experimental details from the appendix to the main body of the paper to ensure that readers can fully understand the experiments without needing to constantly refer to the appendix. This includes more detailed descriptions of the specific evaluation descriptions and results for each task (forecasting, imputation, and super-resolution).
>
> 3. **Limitations of This Work**:
>
>     We have moved the discussion of the method’s limitations from the appendix to the Discussion section in the main body of the paper. We agree that discussing limitations is an important part of the main narrative and should not be relegated to the appendix.
>
> 4. **Minor Changes**:
>
>     - Citations: We have double-checked and corrected the citations throughout the manuscript to ensure consistent usage of \citep and \cite according to the correct context.
>
>     - Font Size in Tables: We have attempted to adjust the font size of the tables to make it a little bigger to improve readability, especially for printed versions. However, due to the template and the condensed information in the tables, the font might still be smaller than the normal size.
>
>     - Table Formatting (Colors and Annotations): Thanks for asking the formatting issue. We used the red bold and blue underscore format by following the prior works such as iTransformer (Liu et al., 2023), FreTS (Yi et al., 2023), TimesNet (Wu et al., 2022), etc.

---

> > ### Comment · Reviewer_Du9K · 2024-10-22
> > **Thank you for considering my feedback**
> >
> > The revision sufficiently addressed my concerns

---

### Review · Reviewer_xiVg · 2024-08-13

**Summary Of Contributions:**

This paper presents the AdaWaveNet by utilizing the decomposition and wavelet network for time series analyses. AdaWaveNet focuses on the non-stationary problem in time series, which is widely evaluated in three different tasks (a new super-resolution task) and compared with extensive baselines.

**Audience:**

Yes

**Claims And Evidence:**

Yes

**Requested Changes:**

They need to clarify the relation between wavelet analysis and non-stationary time series and the special challenges in super-resolution task.

**Strengths And Weaknesses:**

## Strengths
-	This paper is well-written and clear.
-	The proposed method is reasonable, where adopting wavelet analysis is a good idea.
-	Extensive experiments on diverse tasks are included.
-	The showcase visualization in the Appendix is intuitive and appreciated.

## Weaknesses

1.	I think this paper does not provide a clear connection between wavelet analysis and non-stationary time series. Why using wavelet analysis can help the model tackle non-stationary problems? More discussions are required.
2.	To some extent, the super-resolution task is a special imputation problem. Specifically, it can be viewed as imputing the interval between low-frequency sampled time points. More descriptions of the unique challenges in super-resolution are expected.
3.	In Section 4.1, the authors mentioned “It is often calculated using a moving average, which smooths out short-term fluctuations to highlight longer-term trends.” I think a citation to Autoformer (Wu et al., 2021) is needed since the moving average decomposition is proposed in that paper.

---

> ### Author Response · Authors · 2024-10-11
> **Thank you for the insightful comment**
>
> Thank you for your thorough review and valuable feedback on our paper. We have carefully addressed your comments:
>
> 1. **Clarifying the Connection Between Wavelet Analysis and Non-Stationary Time Series**:
>
>     We appreciate your insightful comment regarding the connection between wavelet analysis and non-stationary time series. To address this, we have expanded the background section (Section 3) in the manuscript to explicitly clarify why wavelet analysis is particularly suited for handling non-stationarity. Specifically:
>     - Wavelet analysis provides multi-scale decomposition: Non-stationary time series exhibit changing statistical properties over time, including changing trends, seasonality, and frequencies. Wavelet transforms decompose signals into both time and frequency components, which allows the model to adaptively focus on different scale. This capability is helpful for capturing the dynamics and changes in frequency that are common in non-stationary data.
>     - The adaptive wavelet transformation in AdaWaveNet further enhances this ability by allowing the wavelet coefficients to be learned dynamically based on the data, rather than being fixed or pre-determined. This adaptability is designed to better handle non-stationarity, as it allows the model to adjust to time-varying patterns in the data without relying on a static set of assumptions.
>
> 2. **Clarifying the Special Challenges in Super-Resolution Task**:
>
>     We agree with the reviewer’s observation that the super-resolution task can be viewed as a special case of imputation. However, we would like to emphasize the unique challenges posed by super-resolution in time series analysis: Unlike traditional imputation, which typically focuses on filling missing data points, super-resolution aims to reconstruct higher-frequency data from a lower-frequency sampling. This requires the model not only to fill gaps but also to generate higher-resolution patterns that may not be directly observable from the lower-resolution data. This information is now clarified in the Section 5.3 in the manuscript.
>
> 3. **Citation to Autoformer for Moving Average Decomposition**:
>
>     Thank you for bringing this to our attention. We have now included the appropriate citation to Autoformer (Wu et al., 2021) in Section 4.1 when discussing the moving average decomposition, as it aligns with the method mentioned.

---

### Review · Reviewer_72rq · 2024-09-27

**Summary Of Contributions:**

The paper introduces the Adaptive Wavelet Network (AdaWaveNet), a novel architecture for time series analysis, addressing the challenge of non-stationary data. The contributions include:

1. Novel Architecture: AdaWaveNet offers an adaptive, multi-scale approach to enhance accuracy and reliability in analyzing non-stationary time series data.

2. Super-Resolution Benchmark: Establishes a new benchmark for super-resolution in time series, improving data quality from under-sampled sequences.

3. Performance Evaluation: Extensive experiments demonstrate AdaWaveNet's superiority over existing methods in forecasting, imputation, and super-resolution tasks, indicating its potential for diverse real-world applications.

**Audience:**

Yes

**Broader Impact Concerns:**

The authors discussed the broader impact in section 7.

**Claims And Evidence:**

Yes

**Requested Changes:**

1. Expand Related Work: It's critical to include a more comprehensive discussion of existing decomposition-based methods, as well as other wavelet analysis works. This will strengthen the context and position of the paper within the field.

2. Clarify Contributions: Clearly differentiate the paper's contributions from existing works. This is important to ensure the novelty and impact of the research are well understood.

3. Non-Stationary Analysis: Conduct and include experimental analysis on how the model addresses non-stationarity. This would secure my recommendation for acceptance by providing evidence of the model's effectiveness in this area.

4. Table 1's score doesn't seem to match PatchTST's original results. However, can the authors clarify where the difference comes from, and can we align it with the PatchTST paper?

**Strengths And Weaknesses:**

Strengths:

1. Innovative Approach: Introduces a novel adaptive wavelet network for handling non-stationary time series data.

2. Comprehensive Evaluation: Provides extensive experiments across multiple datasets and tasks, demonstrating superior performance.

3. Multi-Scale Analysis: Effectively incorporates multi-scale analysis, improving flexibility and robustness.

4. New Benchmark: Establishes a benchmark for super-resolution tasks in time series analysis.

Weaknesses:

1. Lack of Related Works: The paper does not adequately cover existing decomposition-based methods, other than recent FedFormer and CrossFormer discussed already, there are still papers like DRCNN[1], MICN[2], and TEMPO[3] nor does it sufficiently discuss related works in wavelet analysis, which has been used on time series analysis for years.

[1] Zhu, Y., Luo, S., Huang, D., Zheng, W., Su, F., & Hou, B. (2023). DRCNN: decomposing residual convolutional neural networks for time series forecasting. Scientific Reports, 13(1), 15901.

[2] Wang, H., Peng, J., Huang, F., Wang, J., Chen, J., & Xiao, Y. (2023). Micn: Multi-scale local and global context modeling for long-term series forecasting. In The eleventh international conference on learning representations.

[3] Cao, D., Jia, F., Arik, S. O., Pfister, T., Zheng, Y., Ye, W., & Liu, Y. (2023). Tempo: Prompt-based generative pre-trained transformer for time series forecasting. arXiv preprint arXiv:2310.04948.

2. Contribution Clarity: The paper's contributions are weakened by the lack of distinction from previous related works.

3. Non-Stationary Analysis: Although targeting non-stationary time series is highlighted, the paper lacks experimental analysis explaining how and why the model effectively addresses non-stationarity.

---

> ### Author Response · Authors · 2024-10-11
> **Thank you for the insightful comment**
>
> Thank you for your thorough review and valuable feedback on our paper. We have carefully addressed your comments:
>
> 1. **Expand Related Work**:
>
>     We appreciate your suggestion to provide a more comprehensive discussion of decomposition-based methods and wavelet analysis works. In response, we have added a dedicated subsection in the related work section. This subsection now covers a broader range of decomposition-based approaches such as DRCNN, MICN, and TEMPO, which discusses their strengths and limitations, particularly in handling non-stationary time series.
>
> 2. **Clarify Contributions**:
>
>     Thank you for the feedback regarding the differentiation of contributions. In response, we have revised the paper, particularly in the introduction and related work section, to more clearly articulate the unique aspects of AdaWaveNet compared to existing methods in the introduction section. By highlighting the adaptive nature of the wavelet transform and the introduction of a novel grouped linear module, we have clearly differentiated our approach from other wavelet-based and decomposition-based models.
>
> 3. **Non-Stationary Analysis**:
>
>     We have added a comprehensive case study on non-stationary signal analysis in the appendix (Appendix C). This section now includes an experimental analysis using a synthetic non-stationary time series designed to exhibit both trend and frequency changes. By comparing AdaWaveNet with other state-of-the-art models (DLinear, TimesNet, FreTS, PatchTST, and iTransformer), we demonstrate that \textit{AdaWaveNet} achieves the lowest MSE and MAE of effectively capturing both low- and high-frequency components of the signal. Also, we provide visualizations and analysis of outputs from each AdaWaveBlocks, which leverages lifting scheme kernels to decompose signals into multiple scales.
>
> 4. **Table 1's Score for PatchTST**:
>
>     We acknowledge your concern regarding the discrepancy in PatchTST’s results in Table 1. The difference arises from the evaluation setting in our experiments. In the PatchTST paper, the look-back length is fixed at 96, whereas in our setting, the look-back length equals the prediction length (96, 192, 336, 720). The reason of this discrepancy in our study is two-fold:
>
>     -  As also discussed in a prior study DLinear (Zeng2022 https://arxiv.org/pdf/2205.13504 ), the model can better capture the temporal dynamics by maintaining a longer look-back length. We designed it equal for both the look-back and prediction windows, which ensures that the model is provided with sufficient historical context to make reasonable predictions over the chosen forecast length.
>     In the architecture of AdaWaveNet, which focus on multi-scale analysis using wavelets, having a consistent look-back and prediction length enables the model to decompose both the input and output into more comparable temporal components. This design helps better extract and align embeddings across different scales of the data.​
>
>     - We used a random seed of 2024, which could be different from the original implementation (seed 2021). These changes in evaluation settings may causeexplain the variation in results. We have clarified this in the manuscript to avoid any confusion. Upon acceptance of the manuscript, we will also release our implementation code to help with reproducibility.

---

### Decision · Action_Editor_YZX8 · 2024-11-06

**Recommendation:** Accept as is

**Comment:**

This paper proposed a novel method and gave a clear analysis for its utility, where all reviewers agreed about the novelty of the method and clarity of the analysis. The major improvements needed were in organization and better differentiating the work from prior work. It has room for improvement in describing its motivation but none-the-less fits should be accepted.

**Audience:**

Yes.

**Claims And Evidence:**

The claims in the submission are clear and supported by evidence.